# Redundancy-Aware Test-Time Graph Out-of-Distribution Detection

**Yue Hou**[1,2], **He Zhu**[1], **Ruomei Liu**[1], **Yingke Su**[2], **Junran Wu**[1]*, **Ke Xu**[1],

[1]State Key Laboratory of Complex & Critical Software Environment, Beihang University
[2]Shen Yuan Honors College, Beihang University
{hou_yue, roy_zh, rmliu, suyingke, wu_junran, kexu}@buaa.edu.cn

## Abstract

Distributional discrepancy between training and test data can lead models to make inaccurate predictions when encountering out-of-distribution (OOD) samples in real-world applications. Although existing graph OOD detection methods leverage data-centric techniques to extract effective representations, their performance remains compromised by structural redundancy that induces semantic shifts. To address this dilemma, we propose RedOUT, an unsupervised framework that integrates structural entropy into test-time OOD detection for graph classification. Concretely, we introduce the Redundancy-aware Graph Information Bottleneck (ReGIB) and decompose the objective into essential information and irrelevant redundancy. By minimizing structural entropy, the decoupled redundancy is reduced, and theoretically grounded upper and lower bounds are proposed for optimization. Extensive experiments on real-world datasets demonstrate the superior performance of RedOUT on OOD detection. Specifically, our method achieves an average improvement of 6.7%, significantly surpassing the best competitor by 17.3% on the ClinTox/LIPO dataset pair.

## 1 Introduction

Deep learning models have achieved impressive success across a wide range of tasks, yet they can behave unpredictably when faced with out-of-distribution (OOD) data that differ significantly from the training distribution, making high-confidence but incorrect predictions. OOD detection [3, 44, 47] seeks to identify such inputs and is critical for reliable deployment in open-world scenarios. This challenge becomes even more pronounced for graph-structured data due to the non-Euclidean nature and complex topological structures.

Recent advancements [9, 20, 39] in graph OOD detection can be broadly divided into two main categories. **(1) End-to-end methods** [20] aim to train OOD-specific graph neural networks (GNNs) [13, 46] from scratch using only unlabeled ID data. These approaches typically leverage unsupervised objectives such as graph contrastive learning (GCL) to extract discriminative representations that can separate ID and OOD samples during inference ($\triangleright$ Figure 1(a)(i)). **(2) Post-hoc methods** [9, 39] employ well-trained GNNs and fine-tune OOD detectors to refine the predictions or representations at inference time ($\triangleright$ Figure 1(a)(ii)). Specifically, GOODAT [39] stands out by directly optimizing a learnable graph masker on test samples without altering the pre-trained model. This test-time setting with data-centric modification is more practical for removing the dependency on labeled training data or model retraining.

Despite significant advancements in the aforementioned data-centric paradigm, a less-explored challenge persists. While pre-trained models can effectively extract information from ID data, they

---

*Corresponding author

39th Conference on Neural Information Processing Systems (NeurIPS 2025).

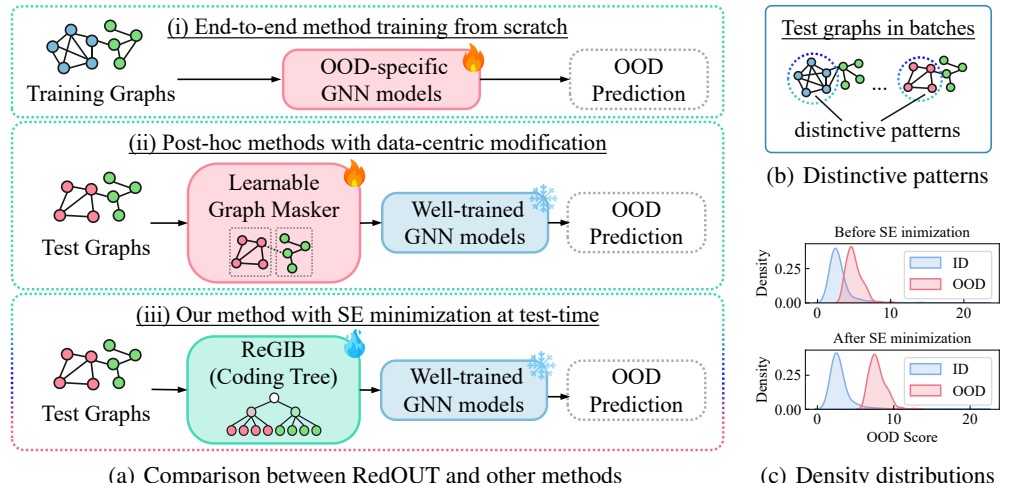

Figure 1: (a) A schema comparison. (b) A toy example of distinctive components within test graphs. (c) Scoring distributions before/after structural entropy (*abbr.* SE) minimization.

lack ground-truth information indicative of the underlying distribution when inferring on both ID and OOD test data, which is often embedded within the graph structure. As illustrated in the toy example in Figure 1(b), graphs in test batches comprise distinctive components (highlighted by dashed circles) and irrelevant structural elements. Effectively capturing these distinctive components can intuitively differentiate OOD samples from ID graphs. However, the pervasive presence of similar irrelevant structural elements hinders current methods from accurately capturing and distinguishing these essential structures between ID and OOD data. Although GOODAT [39] tries to address this concern by utilizing graph maskers to identify subgraphs in test data that are similar to those in ID graphs from training datasets, it fundamentally relies on learnable graph augmentations. These augmentations can inadvertently alter the semantic information of substructures or lead to information loss, thereby compromising the reliability of OOD detection.

To capture the distinctive information while preserving structural semantics, we first decompose the information within graph into essentiality and redundancy, leveraged by the graph information bottleneck [45]. Structural entropy, which provides a hierarchical abstraction to measure structural complexity [15], further inspires us to utilize the technology to eliminate decoupled redundant information. We compute the scoring distributions of graph representations before and after structural entropy minimization on the AIDS/DHFR dataset pair (with AIDS as ID dataset and DHFR as OOD dataset), as shown in Figure 1(c). From score density plots, we observe that after structural entropy minimization, OOD scores exhibit smaller variance and a decrease in the overlap between OOD and ID samples. This intuitively demonstrates that by removing redundancy, the more distinctive parts of graphs are retained, enabling more effective differentiation of the distributions.

In this paper, we propose an innovative redundancy-aware test-time graph OOD detection framework, termed **RedOUT**, aiming at endowing well-trained models with the ability to extract the essential structural information from test graphs effectively. Concretely, we introduce the Redundancy-aware Graph Information Bottleneck (ReGIB) and decompose the objective into distinctive essential view and irrelevant redundancy. By minimizing structural entropy, coding tree is constructed to instantiate the essential view, effectively removing redundancy. To make the overall objectives tractable, we establish upper and lower bounds for optimizing essential and redundant information. A comparison between RedOUT and existing methods is illustrated in Figure 1(a). It is important to note that structural entropy minimization serves as a preprocessing step, utilizing an approximation algorithm that does not require additional learning. Leveraging hierarchical representation learning based on the coding tree, we calibrate OOD scores to get better predictions without modifying the parameters of pre-trained models. Extensive experiments on real-world datasets demonstrate the superiority of RedOUT against state-of-the-art (SOTA) baselines. The contributions of this work are as follows:

• We propose a novel framework for test-time unsupervised graph OOD detection, termed RedOUT. To the best of our knowledge, this is the first trial to endow trained models with the ability to capture the essential structural information of graphs during test-time.

- We introduce the ReGIB to decouple the essential and redundant information within graphs, where coding tree with minimized structural entropy is instantiated as essential view. We further introduce the upper and lower bounds for optimizing the ReGIB objectives.
- Extensive experiments validate the effectiveness of RedOUT, demonstrating the superior performance over SOTA baselines in unsupervised OOD detection.

## 2  Notations and Preliminaries

Before formulating the research problem, we first provide some necessary notations. Let $G = (\mathcal{V}, \mathcal{E}, \mathbf{X})$ represent a graph, where $\mathcal{V}$ is the set of nodes and $\mathcal{E}$ is the set of edges. The node features are represented by the feature matrix $\mathbf{X} \in \mathbb{R}^{n \times d_f}$, where $n = |\mathcal{V}|$ is the number of nodes and $d_f$ is the feature dimension. The structure information can also be described by an adjacency matrix $\mathbf{A} \in \mathbb{R}^{n \times n}$, so a graph can be alternatively represented by $G = (\mathbf{A}, \mathbf{X})$. Notation with its description can be found in Appendix A.

**Unsupervised Pre-training with Contrastive Loss.** In the general graph contrastive learning paradigm for graph classification, two augmented graphs are generated using different graph augmentation operators. Subsequently, representations are generated using a GNN encoder, and further mapped into an embedding space by a shared projection head for contrastive learning. A typical graph contrastive loss, InfoNCE [5, 53], treats the same graph $G_i$ in different views $G_i^\alpha$ and $G_i^\beta$ as positive pairs and other nodes as negative pairs. The graph contrastive learning loss $\mathcal{L}_{Cl}$ can be formulated as:

$$\mathcal{L}_{Cl}(G^\alpha, G^\beta) = -\frac{1}{N} \sum_{i=1}^{N} \log \frac{e^{sim(\mathbf{Z}_i^\alpha, \mathbf{Z}_i^\beta)/\tau}}{\sum_{j=1, j \neq i}^{N} e^{sim(\mathbf{Z}_i^\alpha, \mathbf{Z}_j^\alpha)/\tau}}, \tag{1}$$

where $\mathbf{Z}_i^\alpha$ and $\mathbf{Z}_i^\beta$ are graph-level representations on $G_i^\alpha$ and $G_i^\beta$, $N$ denotes the batch size, $\tau$ is the temperature coefficient, and $sim(\cdot, \cdot)$ stands for cosine similarity function.

In this study, inspired by GOOD-D[20], we employ 5 layers of GIN [46] as the backbone and adopt a perturbation-free augmentation strategy to construct view $G^\gamma = (\mathbf{A}, \mathbf{P})$, where $\mathbf{P}$ is formed by concatenating $\mathbf{p}_i = [\mathbf{p}_i^{(rw)} || \mathbf{p}_i^{(lp)}]$. Specifically, the random walk diffusion encoding $\mathbf{p}_i^{(rw)} = [\text{RW}_{ii}, \text{RW}_{ii}^2, \cdots, \text{RW}_{ii}^r] \in \mathbb{R}^r$, where $\text{RW} = \mathbf{A}\mathbf{D}^{-1}$ is the random walk transition matrix, and $\mathbf{D}$ is the diagonal degree matrix. The Laplacian positional encoding $\mathbf{p}_i^{(lp)} = [\mathbf{I} - \mathbf{D}^{-\frac{1}{2}}\mathbf{A}\mathbf{D}^{-\frac{1}{2}}]_{ii}$. With the original graph $G$ as the basic view, loss $\mathcal{L}_{Cl}(G, G^\gamma)$ is computed to optimize the pre-trained model.

**Test-time Graph-level OOD Detection.** Following GOODAT [39], we consider an unlabeled ID dataset $\mathcal{D}^{id} = \{G^{id}\}_N$ where graphs are sampled from distribution $\mathbb{P}^{id}$ and an OOD dataset $\mathcal{D}^{ood} = \{G^{ood}\}_{N'}$ sampled from a different distribution $\mathbb{P}^{ood}$. Given a graph $G$ from $\mathcal{D}_{test}^{id} \cup \mathcal{D}_{test}^{ood}$, test-time graph OOD detection aims to detect whether $G$ originates from $\mathbb{P}^{id}$ or $\mathbb{P}^{ood}$ utilizing a GNN $f$ pre-trained on ID graphs $\mathcal{D}_{train}^{id} \subset \mathcal{D}^{id}$. Specifically, the objective is to learn an OOD detector $D(\cdot, \cdot)$ that assigns an OOD detection score $s = D(f, G)$, with a higher $s$ indicating a greater probability that $G$ is from $\mathbb{P}^{ood}$ (note that $\mathcal{D}_{test}^{id} \cap \mathcal{D}_{train}^{id} = \emptyset$, $\mathcal{D}_{test}^{id} \subset \mathcal{D}^{id}$, and $\mathcal{D}_{test}^{ood} \subset \mathcal{D}^{ood}$).

**Structural Entropy.** Structural entropy is initially proposed [15] to measure the uncertainty of graph structure, revealing the essential structure of a graph. The structural entropy of a given graph $G = \{\mathcal{V}, \mathcal{E}, \mathbf{X}\}$ on its coding tree $T$ is defined as:

$$\mathcal{H}^T(G) = -\sum_{v_\tau \in T} \frac{g_{v_\tau}}{vol(\mathcal{V})} \log \frac{vol(v_\tau)}{vol(v_\tau^+)}, \tag{2}$$

where $v_\tau$ is a node in $T$ except for root node and also stands for a subset $\mathcal{V}_\tau \in \mathcal{V}$, $g_{v_\tau}$ is the number of edges connecting nodes in and outside $\mathcal{V}_\tau$, $v_\tau^+$ is the immediate predecessor of $v_\tau$ and $vol(v_\tau)$, $vol(v_\tau^+)$ and $vol(\mathcal{V})$ are the sum of degrees of nodes in $v_\tau$, $v_\tau^+$ and $\mathcal{V}$, respectively.

## 3  Methodology

This section elaborates on RedOUT with its framework shown in Figure 2. We first derive the Redundancy-aware Graph Information Bottleneck (ReGIB) ($\triangleright$ Sec. 3.1) by decomposing the objective into essential information and irrelevant redundancy, and introduce their upper and lower bounds. Additionally, we construct coding tree with minimized structural entropy as the essential view and instantiate the ReGIB with tractable bounds for efficient optimization ($\triangleright$ Sec. 3.2).

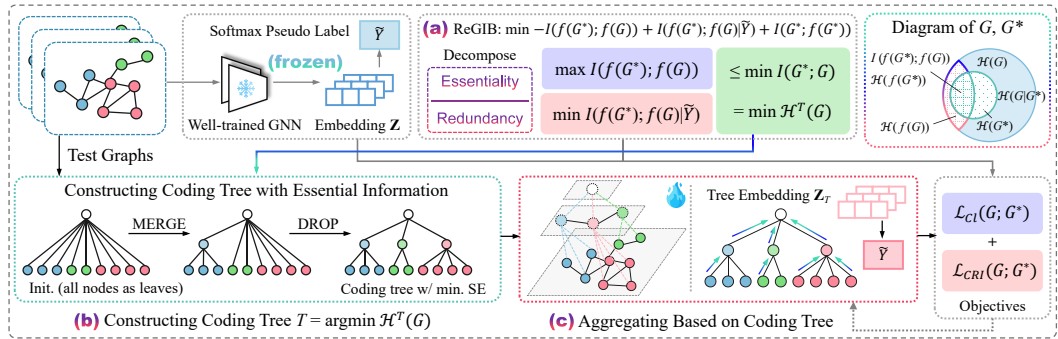

Figure 2: Overview of the overall framework of RedOUT and proposed ReGIB. (a) the ReGIB principle and its bounds. (b) obtained by minimizing structural entropy of graph $G$, coding tree $T$ serves as an instantiation of the essential view. (c) message passing and aggregating on graph are under the guidance of the coding tree.

## 3.1 The Principle of ReGIB

According to the graph information bottleneck (GIB) principle [45], retaining optimal representation on a graph view should involve maximizing mutual information (MI) between the output and ground-truth labels (*i.e.*, $\max I(f(G); Y)$) while reducing mutual information between input and output (*i.e.*, $\min I(G; f(G))$). This can be expressed as follows:

$$\min \text{GIB} \triangleq -I(f(G); Y) + \beta I(G; f(G)), \tag{3}$$

where $\beta$ is the Lagrangian parameter to balance the two terms, $Y$ is the ground-truth labels, and $I(\cdot; \cdot)$ denotes the mutual information between inputs.

As mentioned, ID and OOD graphs can be distinguished based on distinctive structures. In this section, we introduce $G^*$ with essential information of the original graph $G$ in test-time setting for the first time, extending the GIB principle.

Compared to vanilla GIB, the proposed ReGIB accounts for the fact that the model pre-trained solely on ID graphs, struggles to generalize to unseen distribution, leading to the representations containing not only the optimal components but also irrelevant redundancy ($\triangleright$ Figure 3(b)). Moreover, for unsupervised tasks, the predicted label $\tilde{Y}$ is used as a surrogate for

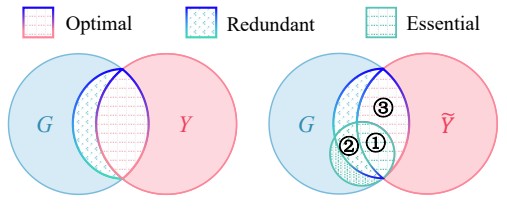

(a) Vanilla GIB      (b) ReGIB

Figure 3: Comparison between the basic GIB and proposed ReGIB. Note ReGIB balances signals ①, ② with optimal $f(G^*)$ to capture the essential information and discard redundancy.

the unknown label distribution, enabling training without access to ground-truth labels $Y$. Thus, by substituting $G^*$ and $\tilde{Y}$ into Eq. (3), we have:

$$\min -I(f(G^*); \tilde{Y}) + \beta I(G^*; f(G^*)). \tag{4}$$

Besides, due to semantic shifts in representations during test-time, the dual information sources ($f(G^*)$ and $\tilde{Y}$) may contain redundany. Therefore, it is crucial to further decouple the dependence among $f(G^*)$, $f(G)$, and $\tilde{Y}$.

**Proposition 1** (Lower Bound of $I(f(G^*); \tilde{Y})$).

$$I(f(G^*); \tilde{Y}) = I(f(G^*); f(G)) + I(f(G^*); \tilde{Y}|f(G)) - I(f(G^*); f(G)|\tilde{Y})$$

$$\geq \underbrace{I(f(G^*); f(G))}_{\text{Eq. (7) and (15)}} - \underbrace{I(f(G^*); f(G)|\tilde{Y})}_{\text{Eq. (8) and (16)}}. \tag{5}$$

Proofs for Proposition 1 is provided in Appendix C.2.

**Definition 3.1** (Redundancy-aware Graph Information Bottleneck). Given a test graph $G$, and the pseudo-label $\tilde{Y}$ predicted by pre-trained $f$, ReGIB aims to capture the distinctive structure $G^*$ with essential information and further learn the optimal representation $f(G^*)$ that satisfies:

$$\min \text{ReGIB} \triangleq -I(f(G^*); f(G)) + I(f(G^*); f(G)|\tilde{Y}) + \beta I(G^*; f(G^*)). \tag{6}$$

**Remark.** Intuitively, the first term $I(f(G^*); f(G))$ is the prediction term, which encourages that essential information to the graph property is preserved. The last two terms $I(f(G^*); f(G)|\tilde{Y})$ and $I(G^*; f(G^*))$ are used for compression, which encourage that irrelevant redundant information is dropped. Since the well-trained model $f$ tends to make accurate predictions on ID graphs, its predictions for OOD graphs, which are unseen during training, are more random. The model struggles to identify distinctive structural information and similar redundant structures. However, with the distinctive patterns of $G^*$, ReGIB can provide ground-truth information to calibrate the model's predictions on test graphs. A further comparison of ReGIB and GIB on an argumentation view is conducted in Appendix C.9.

The key to our objective is how to obtain the distinctive structure $G^*$ and how to acquire the most optimal and essential information based on pseudo-labels while eliminating irrelevant redundancy in representations. Thus, we introduce the lower and upper bounds for Eq. (6).

**Proposition 2** (Lower Bound of $I(f(G^*); f(G))$)**.**

$$I(f(G^*); f(G)) \geq -\mathcal{L}_{Cl}(G^*, G) + \log(N). \tag{7}$$

**Proposition 3** (Upper Bound of $I(f(G^*); f(G)|\tilde{Y})$)**.** For any $\mathbb{Q}(f(G)|\tilde{Y})$, the variational upper bound of conditional mutual information $I(f(G^*); f(G)|\tilde{Y})$ is:

$$I(f(G^*); f(G)|\tilde{Y}) \leq \mathbb{E}_{\mathbb{P}}\left[\log \frac{\mathbb{P}(f(G), f(G^*)|\tilde{Y})}{\mathbb{P}(f(G^*)|\tilde{Y})\mathbb{Q}(f(G)|\tilde{Y})}\right]. \tag{8}$$

Proofs for Proposition 2 and 3 are provided in AppendixC.3 and C.4.

**Redundancy-eliminated Essential Information.** The key to effectively distinguishing between ID and OOD graphs lies in maximizing the elimination of redundancy while preserving essential information. Suppose the optimal view $G^*$ can capture the essential information while eliminating redundancy of graph $G$. Here, we first provide the definition of essential view as follows.

**Definition 3.2.** The essential view with redundancy-eliminated information is supposed to be a distinctive substructure of the given graph.

Based on the above analysis, the redundancy-eliminated view should retain the minimal amount of information while capturing the most distinctive structural patterns. The mutual information between $G$ and $G^*$ can be formulated as:

$$I(G^*; G) = \mathcal{H}(G^*) - \mathcal{H}(G^*|G), \tag{9}$$

where $\mathcal{H}(G^*)$ is the entropy of $G^*$ and $\mathcal{H}(G^*|G)$ is the conditional entropy of $G^*$ conditioned on $G$.

**Theorem 3.3.** The information in $G^*$ is a subset of information in $G$ (i.e., $\mathcal{H}(G^*) \subseteq \mathcal{H}(G)$); thus, we have:

$$\mathcal{H}(G^*|G) = 0. \tag{10}$$

The detailed proof of Theorem 3.3 is shown in Appendix C.5. Here, the mutual information between $G$ and $G^*$ can be rewritten as:

$$I(G^*; G) = \mathcal{H}(G^*). \tag{11}$$

**Proposition 4** (Upper Bound of $I(G^*; f(G^*))$)**.** Given the tuple $(G, G^*, f(G^*))$, the learning procedure follows the Markov Chain $< G \rightarrow G^* \rightarrow f(G^*) >$. Accordingly, to acquire the essential view with essential information, we need to optimize:

$$\min I(G^*; f(G^*)) \leq \min I(G^*; G) = \min \mathcal{H}(G^*). \tag{12}$$

To eliminate redundancy, we introduce structural entropy as a measure of the information content within the hierarchy of the graph. Thus, we argue that the view obtained by minimizing structural entropy of a given graph represents the redundancy-eliminated information, serving as an essential view that retains the graph's distinctive substructure.

To extract optimal and essential representations on $G^*$, we aim to maximize $I(f(G^*); f(G))$ and minimize $I(f(G^*); f(G)|\tilde{Y})$ to obtain the optimal encoder, such that:

$$f^* = \arg\max_f I(f(G^*); f(G)) - I(f(G^*); f(G)|\tilde{Y}). \tag{13}$$

We also analytically provide theoretical guarantee for redundant and essential information as follows.

**Theorem 3.4.** According to Proposition 2, optimizing loss $\mathcal{L}_{Cl}(G^*, G)$ is equivalent to maximizing the mutual information $I(f(G^*); f(G))$, which could lead to the maximization of $I(f(G^*); Y)$.

**Remark.** Theorem 3.4 (See proof in Appendix C.6) reveals that our essential view enables the encoder to preserve more information associated with the ground-truth labels, which will boost the performance of downstream tasks.

**Theorem 3.5** (Bounds of Redundant and Essential Information). Suppose the encoder $f$ is implemented by a GNN as powerful as the 1-WL test. Suppose $\mathcal{G}$ is a countable space and thus $\mathcal{G}'$ is a countable space, where $\cong$ denotes equivalence ($G_1 \cong G_2$ if $G_1, G_2$ cannot be distinguished by the 1-WL test). Define $\mathbb{P}_{\mathcal{G}' \times \mathcal{Y}}(G', Y') = \mathbb{P}_{\mathcal{G} \times \mathcal{Y}}(G \cong G', Y)$ and $\mathcal{T}'(G') = \mathbb{E}_{G \sim \mathbb{P}_G}[\mathcal{T}^*(G)|G \cong G']$ for $G' \in \mathcal{G}$. Then, the optimal $f^*$ and $G^*$ to RedOUT satisfies:

1. $I(G^*; Y) = I(f^*(G^*); Y) \geq I(t'(G'); Y)$,

2. $I(f^*(G); G|Y) \leq \min_{\mathcal{T}} I(t'(G'); G')) - I(t'(G'); Y)$,

where $t^*(G) = G^*, t'(G') \sim \mathcal{T}'(G'), t'^*(G') \sim \mathcal{T}'^*(G'), (G, Y) \sim \mathbb{P}_{\mathcal{G} \times \mathcal{Y}}$ and $(G', Y) \sim \mathbb{P}_{\mathcal{G}' \times \mathcal{Y}}$.

Therefore, the encoder optimized by ReGIB enables encoding a representation that has limited redundant information and more essential information (See proof in Appendix C.7).

## 3.2 ReGIB Principle Instantiation

To optimize ReGIB, we begin by constructing coding tree to instantiate essential information, and then specify the lower and upper bounds defined in Proposition 2, 3 and 4.

**Coding Tree Construction (Instantiation for $G^*$).** For any given test graph, we construct an essential view of the graph with minimal structural entropy, as defined by Eq. (12). Specifically, according to structural information theory [15], the structural entropy of a graph needs to be calculated with the coding tree. Besides the optimal coding tree with minimum structural entropy (*i.e.*, $\min_{\forall T}\{\mathcal{H}^T(G)\}$), a fixed-height coding tree is often preferred for its better representing the fixed natural hierarchy commonly found in real-world networks. Therefore, the $k$-dimensional structural entropy of $G$ is defined on coding tree with fixed height $k$:

$$\mathcal{H}^{(k)}(G) = \min_{\forall T: \text{Height}(T)=k}\{\mathcal{H}^T(G)\}. \tag{14}$$

The total process of generation of a coding tree with fixed height $k$ can be divided into two steps: 1) construction of the full-height binary coding tree, and 2) compression of the binary coding tree to height $k$. Given root node $v_r$ of the coding tree $T$, all original nodes in graph $G = (\mathcal{V}, \mathcal{E})$ are treated as leaf nodes. Correspondingly, based on SEP [42], we design two efficient operators, **MERGE** and **DROP**, to construct a coding tree $T$ with minimum structural entropy. An overview of these operators is shown in Algorithm 1 (in Appendix D).

**Remark.** The coding tree, as a hierarchical abstraction of the original graph structure, can be considered a contrastive view $T = \arg\min \mathcal{H}^T(G)$, which serves as an instantiation of the essential information in graph $G$. By minimizing structural entropy, this essential view $T$ effectively reduces redundant information from the graphs while preserving distinctive structural features, enabling the capture of distinct patterns between ID and OOD samples.

**Representing Learning on Essential Information.** Within the test-time OOD detection setting, the parameters of the pre-trained model are frozen. To maintain the optimal representation of the essential information, we update parameters of the coding tree encoder based on the tree essential view $T$ constructed during preprocessing. Specifically, the encoder is designed to iteratively transfer messages from the bottom to the top. Formally, the $l$-th layer of the encoder can be written as, $\mathbf{x}_v^{(l)} = \text{MLP}^{(l)}\left(\sum_{u \in \mathcal{C}(v)} \mathbf{x}_u^{(l-1)}\right)$, where $\mathbf{x}_v^i$ is the feature of $v$ in the $i$-th layer of coding tree $T$, $\mathbf{x}_v^0$ is the input feature of leaf nodes, and $\mathcal{C}(v)$ refers to the children of $v$. The aggregated node features from the $(l-1)$-th layer of coding tree are used as inputs for the $l$-th layer, continuing the propagation towards the top of the tree. Once the features reach the root node, a readout function is applied to obtain the tree embedding $\mathbf{Z}_T$.

**Instantiation for $I(f(G^*); f(G))$.** Based on the lower bound derived in Eq. (7), the mutual information between the essential view $G$ and basic view $G$ is estimated by:

$$I(f(G^*); f(G)) \doteq -\mathcal{L}_{Cl}(G^*, G). \tag{15}$$

Table 1: OOD detection results in terms of AUC (%, mean ± std). The best and runner-up results are highlighted with **bold** and underline, respectively. A.A. is short for average AUC. A.R. implies the abbreviation of average rank. The results of baselines are derived from the published works, with unreported results denoted by '−'.

| ID dataset
OOD dataset | BZR
COX2 | PTC-MR
MUTAG | AIDS
DHFR | ENZYMES
PROTEIN | IMDB-M
IMDB-B | Tox21
SIDER | FreeSolv
ToxCast | BBBP
BACE | ClinTox
LIPO | Esol
MUV | A.A. | A.R. |
|---|---|---|---|---|---|---|---|---|---|---|---|---|
| *Non-deep Two-step Methods* | | | | | | | | | | | | |
| PK-LOF | 42.22±8.39 | 51.04±6.04 | 50.15±3.29 | 50.47±2.87 | 48.03±2.53 | 51.33±1.81 | 49.16±3.70 | 53.10±2.07 | 50.00±2.17 | 50.82±1.48 | 49.63 | 14.9 |
| PK-OCSVM | 42.55±8.26 | 49.71±6.58 | 50.17±3.30 | 50.46±2.78 | 48.07±2.41 | 51.33±1.81 | 48.82±3.29 | 53.05±2.10 | 50.06±2.19 | 51.00±1.33 | 49.52 | 14.8 |
| PK-iF | 51.46±1.62 | 54.29±4.33 | 51.10±1.43 | 51.67±2.69 | 50.67±2.47 | 49.87±0.82 | 52.28±1.87 | 51.47±1.33 | 50.81±1.10 | 50.85±3.51 | 51.45 | 12.9 |
| WL-LOF | 48.99±6.20 | 53.31±8.98 | 50.77±2.87 | 52.66±2.47 | 52.28±4.50 | 51.92±1.58 | 51.47±4.23 | 52.80±1.91 | 51.29±3.40 | 51.26±1.31 | 51.68 | 12.1 |
| WL-OCSVM | 49.16±4.51 | 53.31±7.57 | 50.98±2.71 | 51.77±2.21 | 51.38±2.39 | 51.08±1.46 | 50.38±3.81 | 52.85±2.00 | 50.77±3.69 | 50.97±1.65 | 51.27 | 13.0 |
| WL-iF | 50.24±2.49 | 51.43±2.02 | 50.10±0.44 | 51.17±2.01 | 51.07±2.25 | 50.25±0.96 | 52.60±2.38 | 50.78±0.75 | 50.41±2.17 | 50.61±1.96 | 50.87 | 14.2 |
| *Deep Two-step Methods* | | | | | | | | | | | | |
| InfoGraph-iF | 63.17±9.74 | 51.43±5.19 | 93.10±1.35 | 60.00±1.83 | 58.73±1.96 | 56.28±0.81 | 56.92±1.69 | 53.68±2.90 | 48.51±1.87 | 54.16±5.14 | 59.60 | 9.9 |
| InfoGraph-MD | 86.14±6.77 | 50.79±8.49 | 69.02±11.67 | 55.25±3.51 | **81.38±1.14** | 59.97±2.06 | 58.05±5.46 | 70.49±4.63 | 48.12±5.72 | 77.57±1.69 | 65.68 | 8.5 |
| GraphCL-iF | 60.00±3.81 | 50.86±4.30 | 92.90±1.21 | 61.33±2.27 | 59.67±1.65 | 56.81±0.97 | 55.55±2.71 | 59.41±3.58 | 47.84±0.92 | 62.12±4.01 | 60.65 | 10.2 |
| GraphCL-MD | 83.64±6.00 | 73.03±2.38 | 93.75±2.13 | 52.87±6.11 | 79.09±2.73 | 58.30±1.52 | 60.31±5.24 | 75.72±1.54 | 51.58±3.64 | 78.73±1.40 | 70.70 | 6.3 |
| *End-to-end Training Methods* | | | | | | | | | | | | |
| OCGIN | 76.66±4.17 | 80.38±6.84 | 86.01±6.59 | 57.65±2.96 | 67.93±3.86 | 46.09±1.66 | 59.60±4.78 | 61.21±8.12 | 49.13±4.13 | 54.04±5.50 | 63.87 | 9.3 |
| GLocalKD | 75.75±5.99 | 70.63±3.54 | 93.67±1.24 | 57.18±2.03 | 78.25±4.35 | 66.28±0.98 | 64.82±3.31 | 73.15±1.26 | 55.71±3.81 | 86.83±2.35 | 72.23 | 6.1 |
| GOOD-D$_{simp}$ | 93.00±3.20 | 78.43±2.67 | 98.91±0.41 | 61.89±2.51 | 79.71±1.19 | 65.30±1.27 | 70.48±2.75 | 81.56±1.97 | 66.13±2.98 | 91.39±0.46 | 78.68 | 3.4 |
| GOOD-D | 94.99±2.25 | 81.21±2.65 | 99.07±0.40 | 61.84±1.94 | 79.94±1.09 | 66.50±1.35 | 80.13±3.43 | 82.91±2.58 | 69.18±3.61 | 91.52±0.70 | 80.73 | 2.4 |
| *Test-time and Data-centric Methods* | | | | | | | | | | | | |
| GTrans | 55.17±5.04 | 62.38±2.36 | 60.12±1.98 | 49.94±5.67 | 51.55±2.90 | 61.67±0.73 | 50.81±3.03 | 64.02±2.10 | 58.54±2.38 | 76.31±3.85 | 59.05 | 10.0 |
| AAGOD$_S$+ | 76.75 | – | – | 66.22 | 59.00 | 64.26 | – | 67.80 | – | – | – | – |
| AAGOD$_L$+ | 76.00 | – | – | 65.89 | 62.70 | 57.59 | – | 57.13 | – | – | – | – |
| GOODAT | 82.16±0.15 | 81.84±0.57 | 96.43±0.25 | 66.29±1.54 | 79.03±0.03 | 68.92±0.01 | 68.83±0.02 | 77.07±0.03 | 62.46±0.54 | 85.91±0.27 | 76.89 | 3.9 |
| RedOUT | **95.06±0.54** | **94.45±1.66** | **99.98±0.16** | **66.75±2.02** | 79.54±0.72 | **71.67±0.50** | **92.97±0.84** | **92.60±0.23** | **86.56±0.76** | **95.00±0.54** | **87.46** | **1.3** |
| Improve | △+0.07 | △+12.61 | △+0.91 | △+0.46 | ▽ | △+2.75 | △+12.84 | △+9.69 | △+17.38 | △+3.48 | △+6.73 | △ |

**Instantiation for** $I(f(G^*); f(G)|\tilde{Y})$**.** To specify the variational upper bound, we treat the similarity between $\mathbf{Z}_T$ and $\mathbf{Z}$ given the predicted label $\tilde{Y} = \arg\max(\text{softmax}(\mathbf{Z}))$ as an approximation of the log-likelihood $\log \mathbb{P}(f(G)|f(G^*), \tilde{Y})$, and set $\mathbb{Q}(f(G)|\tilde{Y}) = \mathbb{E}_{\mathbf{Z}^- \sim \mathbb{P}(f(G)|\tilde{Y})} e^{\text{sim}(\mathbf{Z}^-, \mathbf{Z}_T)}$, where $\mathbf{Z}^-$ are negative samples drawn from the conditional distribution $\mathbb{P}(f(G)|\tilde{Y})$. Thus, $I(f(G^*); f(G) \mid \tilde{Y})$ can be approximately instantiated as (Proof in Appendix C.8):

$$I(f(G^*); f(G)|\tilde{Y}) \doteq \mathcal{L}_{CRI}(G, G^*), \tag{16}$$

where $\mathcal{L}_{CRI}$ is conditional redundancy-eliminated loss, *i.e.*,

$$\mathcal{L}_{CRI}(G, G^*) \triangleq \mathbb{E}_{\tilde{y} \sim \mathbb{P}(\tilde{Y})} \mathbb{E}_{\mathbf{Z}, \mathbf{Z}_T \sim \mathbb{P}(f(G), f(G^*)|\tilde{y})} \left[ sim(\mathbf{Z}, \mathbf{Z}_T) - \log \mathbb{E}_{\mathbf{Z}^- \sim \mathbb{P}(f(G)|\tilde{y})} e^{sim(\mathbf{Z}^-, \mathbf{Z}_T)} \right]. \tag{17}$$

**Optimization.** Regarding objectives in Eq. (15) and (16), the overall optimization objective is:

$$\mathcal{L} = \mathcal{L}_{Cl} + \lambda \mathcal{L}_{CRI}, \tag{18}$$

where $\lambda$ is a trade-off hyperparameter. When implementing OOD detection, we employ this overall loss $\mathcal{L}$ as the OOD detection score.

## 4 Experiment

In this section, we empirically evaluate the effectiveness of the proposed RedOUT.[2] Detailed settings and additional results can be found in Appendix E.

**Datasets.** For OOD detection, we employ 10 pairs of datasets from two mainstream graph data benchmarks (i.e., TUDataset [24] and OGB [11]) following GOOD-D [20]. We also conduct experiments on anomaly detection settings, where the samples in minority class or real anomalous class are viewed as anomalies. Further details are shown in Appendix E.1.

**Baselines.** We compare RedOUT with 18 competing baseline methods, including 6 graph kernel based methods [32, 38], 4 GCL [20, 34, 49] based methods, 4 end-to-end training methods [21, 51], 1 test-time training methods [12], and 3 data-centric OOD detection methods [9, 39]. More details about implementation are provided in Appendix E.2.

**Evaluation and Implementation.** We evaluate RedOUT with a popular OOD detection metric, *i.e.*, the area under receiver operating characteristic Curve (AUC). Higher AUC values indicate better performance. The reported results are the mean performance with standard deviation after 5 runs.

---

[2]The code of RedOUT is available at: `https://github.com/name-is-what/RedOUT`.

## 4.1 Performance on OOD Detection

In this section, we compare our proposed methods with 18 competing methods in OOD detection tasks. From the comparison results in Table 1, we observe that RedOUT achieves superior performance improvements, which outperforms other baselines on 9 out of 10 dataset pairs and ranks first on average among all baselines with an average rank (A.R.) of 1.3. Concretely, RedOUT achieves an average AUC of 87.46, outperforming all compared methods by 6.7% over the second-best approach GOOD-D [20]. Moreover, RedOUT delivers nearly a 10% performance gain on 4 pairs of molecular datasets. Notably, on the ClinTox/LIPO dataset pair, RedOUT surpasses the best competitor by 17.3%. These findings highlight the superiority of RedOUT in OOD detection tasks, demonstrating its strong capability to capture essential information in graph data.

We also observe that RedOUT does not achieve the absolute best results on the IMDB-B/IMDB-M dataset pair. We consider this phenomenon to be within our expectation, as for molecular graphs, semantic information is directly manifested in their structural composition (*e.g.*, molecular functional groups), thereby enhancing effectiveness. In contrast, for social networks such as IMDB-B and IMDB-M, which originate from the same data source and differ only in labels, the inherent semantic information is similarly reflected structurally, making it challenging to distinguish based solely on structure. A further explanation is provided through a case study in Sec. 4.5.

**Time Complexity Analysis.** Given a graph $G = (\mathcal{V}, \mathcal{E})$, the time complexity of coding tree construction is $O(h(|\mathcal{E}| \log |\mathcal{V}| + |\mathcal{V}|))$, in which $h$ is the height of coding tree $T$ after the first step. In general, the coding tree $T$ tends to be balanced in the process of structural entropy minimization, thus, $h$ will be around $\log |\mathcal{V}|$. Furthermore, a graph generally has more edges than nodes, *i.e.*, $|\mathcal{E}| \gg |\mathcal{V}|$.

To evaluate the efficiency of RedOUT, we plot the runtime and memory usage at peak time on Erdős-Rényi graphs with $|\mathcal{E}| = 2|\mathcal{V}|$ shown in Figure 4. The results illustrate that from smaller scales to the OGB datasets (*e.g.*, ogbg-code2, with an average count of around 100 nodes), the runtime and memory usage scale up nearly linearly with $|\mathcal{V}|$, which is consistent with the theoretical analysis above. RedOUT remains efficient even for graph sizes that exceed those of OGB datasets. We also compared the time consumption in Appendix E.6.

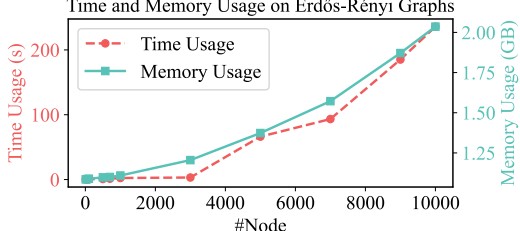

Figure 4: Scalability of coding tree construction on Erdős-Rényi graphs with $|\mathcal{E}| = 2|\mathcal{V}|$.

## 4.2 Performance on Anomaly Detection

To investigate if RedOUT can generalize to the anomaly detection setting [21, 51], we conduct experiments on 5 datasets following the benchmark in GOOD-D [20]. From the results shown in Table 2, we find that RedOUT shows significant performance improvements compared to other baselines, owing to the strong capability to capture essential patterns. More experiments on other 5 datasets for anomaly detection can be found in Appendix E.8.

Table 2: Anomaly detection results in terms of AUC (%, mean $\pm$ std). The best and runner-up results are highlighted with **bold** and underline, respectively.

| Dataset | ENZYMES | DHFR | BZR | IMDB-B | REDDIT-B |
|---------|---------|------|-----|--------|----------|
| PK-OCSVM | 53.67±2.66 | 47.91±3.76 | 46.85±5.31 | 50.75±3.10 | 45.68±2.24 |
| PK-iF | 51.30±2.01 | 52.11±3.96 | 55.32±6.18 | 50.80±3.17 | 46.72±3.42 |
| WL-OCSVM | 55.24±2.66 | 50.24±3.13 | 50.56±5.87 | 54.08±5.19 | 49.31±2.33 |
| WL-iF | 51.60±3.81 | 50.29±2.77 | 52.46±3.30 | 50.20±0.40 | 48.26±0.32 |
| GraphCL-iF | 53.60±4.88 | 51.10±2.35 | 60.24±5.37 | 56.50±4.90 | 71.80±4.38 |
| OCGIN | 58.75±5.98 | 49.23±3.05 | 65.91±1.47 | 60.19±8.90 | 75.93±8.65 |
| GLocalKD | 61.39±8.81 | 56.71±3.57 | 69.42±7.78 | 52.09±3.41 | 77.85±2.62 |
| GOOD-D$_{simp}$ | 61.23±4.58 | 62.71±3.38 | 74.48±4.91 | 65.49±1.06 | 87.87±1.38 |
| GOOD-D | 63.90±3.69 | 62.67±3.11 | 75.16±5.15 | 65.88±0.75 | 88.67±1.24 |
| GTrans | 38.02±6.24 | 61.15±2.87 | 51.97±8.15 | 45.34±3.75 | 69.71±2.21 |
| GOODAT | 52.33±4.74 | 61.52±2.86 | 64.77±3.87 | 65.46±4.34 | 80.31±0.85 |
| RedOUT | **77.64±5.64** | **65.51±4.95** | **89.62±4.72** | **67.48±0.59** | **89.81±2.54** |

## 4.3 Ablation Study

In this section, we conduct ablation experiments on all OOD detection datasets to analyze the effectiveness of two variants by separately removing $\mathcal{L}_{Cl}$ and $\mathcal{L}_{CRI}$. Results are presented in Table 3. Ablating $\mathcal{L}_{Cl}$ prevents the pre-trained model from obtaining a new optimal representation for calibrating the OOD score. In contrast, ablating $\mathcal{L}_{CRI}$ impairs the removal of irrelevant redundant information. This further demonstrates the effectiveness of ReGIB in disentangling essential and redundant information. Concretely, we witness RedOUT surpassing both variants, which provides insights into the effectiveness of the proposed losses and demonstrates their importance in achieving better performance for capturing essential information.

Table 3: Ablation study results of RedOUT and its variants in terms of AUC (%, mean ± std). The best and runner-up results are highlighted with **bold** and underline, respectively.

| $\mathcal{L}_{Cl}$ | $\mathcal{L}_{CRI}$ | BZR | PTC-MR | AIDS | ENZYMES | IMDB-M | Tox21 | FreeSolv | BBBP | ClinTox | Esol |
|---|---|---|---|---|---|---|---|---|---|---|---|
| | | COX2 | MUTAG | DHFR | PROTEIN | IMDB-B | SIDER | ToxCast | BACE | LIPO | MUV |
| ✗ | ✓ | 93.04±1.83 | 86.38±4.72 | 98.79±0.30 | 58.76±3.02 | 73.28±2.13 | 65.06±1.49 | 87.63±4.41 | 86.38±1.54 | 76.75±3.03 | 91.61±2.32 |
| ✓ | ✗ | 92.73±1.71 | 89.40±2.84 | 98.77±0.16 | 58.73±2.81 | 72.67±2.13 | 67.05±1.35 | 88.66±4.40 | 85.99±1.69 | 76.90±2.43 | 91.81±1.77 |
| ✓ | ✓ | **95.06±0.54** | **94.45±1.66** | **99.98±0.16** | **66.75±2.02** | **79.54±0.72** | **71.67±0.50** | **92.97±0.84** | **92.60±0.23** | **86.56±0.76** | **95.00±0.54** |

## 4.4 Parameter Study

**The Height $k$ of Graph's Natural Hierarchy.** Here, we delve deeper into the effect of the height $k$ on the graph's natural hierarchy. The specific performance of RedOUT under each height $k$, ranging from 2 to 5, on OOD detection is shown in Figure 5. We can observe that the optimal height $k$ with the highest accuracy varies among datasets. We also conducted experiments on the impact of coding tree height on anomaly detection datasets, as detailed in Appendix E.9.

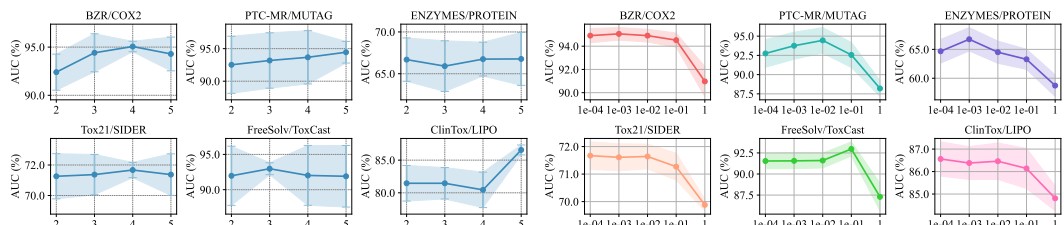

Figure 5: The natural hierarchy of graph.  Figure 6: The sensitivity of hyperparameter $\lambda$.

**Sensitivity Analysis of $\lambda$.** To analyze the sensitivity of $\lambda$ for RedOUT, we alter the value from 1e-04 to 1. The AUC w.r.t different selections of $\lambda$ is plotted in Figure 6. Results demonstrate the performance is sensitive to changes in $\lambda$ and contains a reasonable range across different datasets.

## 4.5 Visualization and Case Study

**Visualization.** The embeddings learned by Red-OUT on AIDS/DHFR are visualized using t-SNE [37] in Figure 7(a)-(c). The representations $\mathbf{Z}$ from the pre-trained model tend to blend ID and OOD graphs. The limitation of GOO-DAT [39] lies in the relatively small representation space, which results in over-compression of representations and consequently diminishes the discriminability between ID and OOD samples. In contrast, the representation gap in $\mathbf{Z}_T$ is most prominent, highlighting its superior effectiveness in capturing essential structures.

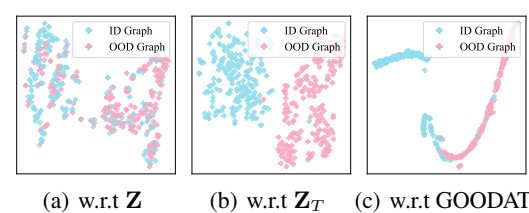

(a) w.r.t $\mathbf{Z}$    (b) w.r.t $\mathbf{Z}_T$    (c) w.r.t GOODAT

Figure 7: T-SNE visualization of embeddings, where $\mathbf{Z}$ is from the pre-trained model, and $\mathbf{Z}_T$ is from the coding tree encoder.

**Case Study.** To further investigate the suboptimal performance of RedOUT on social networks, we visualize the essential structures extracted on IMDB-B and IMDB-M, as shown in Figure 8(a)-(b). These datasets share similar structural characteristics (*e.g.*, star-shaped and mesh-like patterns) but differ in the nature of classification task (binary versus multi-class). The similarity in structural semantics suggests that the intrinsic information captured by the coding tree may not sufficiently distinguish these datasets from a structural standpoint, consistent with the results in Table 1. In contrast, Figure 8(c)-(d) illustrates the extracted essential structures on PTC-MR and MUTAG, highlighting RedOUT's capability in molecular datasets where structural differences are more pronounced and crucial for OOD detection. Additional case studies and analyses are provided in Appendix E.10.

# 5 Related Work

**Graph Out-of-distribution Detection.** Graph OOD detection aims to distinguish OOD graphs from ID ones to address the excessive confidence predictions. Lots of existing methods [54, 16, 17] focus

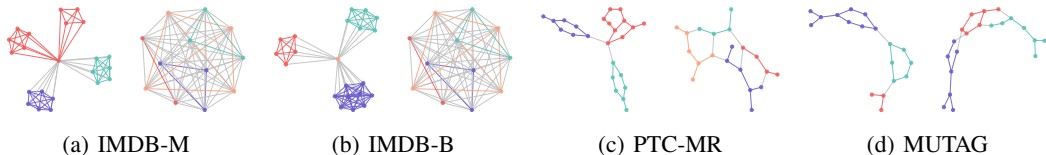

(a) IMDB-M         (b) IMDB-B         (c) PTC-MR         (d) MUTAG

Figure 8: Visualization of the essential structure based on the coding tree preserved by RedOUT on IMDB-B/IMDB-M and PTC-MR/MUTAG dataset pairs.

on improving the generalization ability of GNNs for specific downstream tasks like node classification through supervised learning, rather than identifying OOD samples. Other advancements [9, 20, 39] focus on training OOD-specific GNN models or enable well-trained models in a post-hoc manner with training or test samples. In this work, we provide a novel perspective for identifying OOD graphs at test-time by focusing on essential information based on structural entropy.

**Structural Entropy.** Structural entropy [15], an extension of Shannon entropy [31], quantifies system uncertainty by measuring the complexity of graph structures through the coding tree. Structural entropy has been widely applied in various domains [50, 42, 48, 41, 40], such as dynamic network analysis [19], hierarchical community detection [43], and graph structure learning [55]. In our work, we apply structural entropy to capture the distinctive structure with essential information on test graphs for test-time OOD detection.

**Discussion and Comparison with Related Methods.** Here, we elucidate the association between our proposed RedOUT and two prominent methods, namely GOODAT [39] and SEGO [10].

- **GOODAT.** GOODAT [39] first introduced the setting of test-time OOD detection, where a plug-and-play graph masker is trained to decompose the test-time graph into two subgraphs. Although GOODAT is built upon the GIB principle, it does not consider the redundancy in graph structures and cannot guarantee that the subgraph decomposition preserves semantic information. In contrast, RedOUT is the first to theoretically decouple GIB into redundant and essential components, and improves OOD detection for pre-trained models by explicitly removing redundancy at test-time.
- **SEGO.** SEGO [10] stands out as a pioneering work specifically crafted for unsupervised OOD detection, achieving promising results through a pre-hoc operation that minimizes structural entropy. It is noteworthy that SEGO is a fully end-to-end method trained from scratch, which differs from the setting considered in this paper. Although SEGO also follows structural information principles, it does not investigate what constitutes redundancy or how it should be removed. In contrast, our approach is the first to isolate redundancy from the perspective of GIB and extract the essential structural information.

Further comparisons and analysis[3] can be found in Appendix E.5 and E.6.

## 6    Conclusion

In this paper, we present a redundancy-aware test-time OOD detection framework, termed **RedOUT**, aiming at endowing well-trained GNN models with the ability to extract essential information for the first time. Concretely, we propose the Redundancy-aware Graph Information Bottleneck and decompose the objective into essential information and irrelevant redundancy. By minimizing structural entropy, coding tree is constructed to instantiate the essential view, and tractable bounds are introduced for efficient optimization. Extensive experiments on real-world datasets demonstrate the superior performance of RedOUT over state-of-the-art baselines in unsupervised OOD detection. One limitation is that we mainly consider the graph-level tasks, and leave extending our method to the node-level test-time OOD detection for future explorations.

## Acknowledgements

This work has been supported by CCSE project (CCSE-2024ZX-09).

---

[3]Since SEGO adopts a data-centric preprocessing strategy to assist OOD-specific training, it does not fall under the category of native end-to-end methods. Therefore, we compare it separately with RedOUT in Table 8.

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

# Technical Appendices and Supplementary Material

## A Notation Table

As an expansion of the notations in our work, we summarize the frequently used notations in Table 4.

## B Related Work

**Graph Out-of-distribution Detection.** Graph OOD detection aims to distinguish OOD graphs from ID ones to address the excessive confidence predictions. Lots of existing methods [54, 16, 17] focus on improving the generalization ability of GNNs for specific downstream tasks like node classification through supervised learning, rather than identifying OOD samples. Other advancements [9, 20, 39] focus on training OOD-specific GNN models or enable well-trained models in a post-hoc manner with training or test samples. In this work, we provide a novel perspective for identifying OOD graphs at test-time by focusing on essential information based on structural entropy.

Table 4: The most frequently used notations in this paper.

| Notations | Descriptions |
| --- | --- |
| $G = (\mathcal{V}, \mathcal{E}, \mathbf{X})$ | Graph with the node set $\mathcal{V}$ and edge set $\mathcal{E}$ |
| $\mathcal{V}$ | The set of nodes in the graph |
| $\mathcal{E}$ | The set of edges in the graph |
| $\mathbf{X}$ | The feature matrix |
| $\mathbb{P}^{id}, \mathbb{P}^{ood}$ | The distribution where the graphs are sampled from |
| $Y \in \{0, 1\}$ | The label of test graph |
| $\hat{Y}$ | The predicted label of test graph |
| $\mathbf{A}$ | The adjacency matrix of the graph |
| $G = (\mathbf{A}, \mathbf{X})$ | The original graph as the basic view |
| $G^{\gamma} = (\mathbf{A}, \mathbf{P})$ | Perturbation-free augmentation on $G$ |
| $\mathcal{H}$ | The structural entropy |
| $T$ | Tree essential view of the graph |
| $k$ | The coding tree height |
| $\mathbf{Z}$ | Graph embedding from the well-trained GNN model |
| $\mathbf{Z}_T$ | Tree embedding |
| $\mathcal{L}, \mathcal{L}_{Cl}, \mathcal{L}_{CRI}$ | Overall loss, contrastive loss and conditional redundancy-eliminated loss |
| $\lambda$ | Hyperparameter for loss trade-off |

**Structural Entropy.** Structural entropy [15], an extension of Shannon entropy [31], quantifies system uncertainty by measuring the complexity of graph structures through the coding tree. Structural entropy has been widely applied in various domains [50, 42, 48, 41, 40], such as dynamic network analysis [19], hierarchical community detection [43], and graph structure learning [55]. In our work, we apply structural entropy to capture the distinctive structure with essential information on test graphs for test-time OOD detection.

**Discussion and Comparison with Related Methods.** Here, we elucidate the association between our proposed RedOUT and two prominent methods, namely GOODAT [39] and SEGO [10].

- **GOODAT.** GOODAT [39] first introduced the setting of test-time OOD detection, where a plug-and-play graph masker is trained to decompose the test-time graph into two subgraphs. Although GOODAT is built upon the GIB principle, it does not consider the redundancy in graph structures and cannot guarantee that the subgraph decomposition preserves semantic information. In contrast, RedOUT is the first to theoretically decouple GIB into redundant and essential components, and improves OOD detection for pre-trained models by explicitly removing redundancy at test-time.

- **SEGO.** SEGO [10] stands out as a pioneering work specifically crafted for unsupervised OOD detection, achieving promising results through a pre-hoc operation that minimizes structural entropy. It is noteworthy that SEGO is a fully end-to-end method trained from scratch, which differs from the setting considered in this paper. Although SEGO also follows structural information principles, it does not investigate what constitutes redundancy or how it should be removed. In contrast, our approach is the first to isolate redundancy from the perspective of GIB and extract the essential structural information.

Further comparisons and analysis[4] can be found in Appendix E.5 and E.6.

## C  Theoretical Justification

### C.1  Preliminaries for Mutual Information

**Definition C.1** (Informational Divergence). The informational divergence (also called relative entropy or Kullback-Leibler distance) between two probability distributions $p$ and $q$ on a finite space $\mathcal{X}$ (*i.e.*,

---

[4]Since SEGO adopts a data-centric preprocessing strategy to assist OOD-specific training, it does not fall under the category of native end-to-end methods. Therefore, we compare it separately with RedOUT in Table 8.

a common alphabet) is defined as

$$D_{\mathrm{KL}}(p||q) = \sum_{x \in \mathcal{X}} p(x) \log \frac{p(x)}{q(x)} = \mathbb{E}_p \big[ \log \frac{p(X)}{q(X)} \big].\tag{19}$$

**Remark C.1.** $D_{\mathrm{KL}}(p||q)$ measures the *distance* between $p$ and $q$. However, $D(\cdot||\cdot))$ is not a true metric, and it does not satisfy the triangular inequality. $D_{\mathrm{KL}}(p||q)$ is non-negative and zero if and only if $p = q$.

**Definition C.2** (Mutual Information). Given two discrete random variables $X$ and $Y$, the mutual information (MI) $I(X;Y)$ is the relative entropy between the joint distribution $p(x,y)$ and the product of the marginal distributions $p(x)p(y)$, namely,

$$\begin{aligned} I(X;Y) &= D_{\mathrm{KL}}(p(x,y)||p(x)p(y)) \\ &= \sum_{x \in X, y \in Y} p(x,y) \log \big( \frac{p(x,y)}{p(x)p(y)} \big) \\ &= \sum_{x \in X, y \in Y} p(x,y) \log \big( \frac{p(x|y)}{p(x)} \big). \end{aligned}\tag{20}$$

**Remark C.2.** $I(X;Y)$ is symmetrical in $X$ and $Y$, *i.e.*, $I(X;Y) = \mathcal{H}(X) - \mathcal{H}(X|Y) = \mathcal{H}(Y) - \mathcal{H}(Y|X) = I(Y;X)$.

## C.2 Proof for Proposition C.3

**Proposition C.3** (Lower Bound of $I(f(G^*); \tilde{Y})$).

$$I(f(G^*); \tilde{Y}) \geq I(f(G^*); f(G)) - I(f(G^*); f(G)|\tilde{Y}).\tag{21}$$

*Proof.*

$$I(f(G^*); \tilde{Y}) = I(f(G^*); f(G)) + I(f(G^*); \tilde{Y}|f(G)) - I(f(G^*); f(G)|\tilde{Y}).\tag{22}$$

Since mutual information is non-negative, *i.e.*,

$$I(f(G^*); \tilde{Y}|f(G)) \geq 0.\tag{23}$$

This implies that:

$$I(f(G^*); \tilde{Y}) \geq I(f(G^*); f(G)) - I(f(G^*); f(G)|\tilde{Y}).\tag{24}$$

$\square$

## C.3 Proof for Proposition C.4

To begin with, we revisit the graph contrastive learning loss $\mathcal{L}_{Cl}$, formally expressed as:

$$\mathcal{L}_{Cl}(G^\alpha, G^\beta) = -\frac{1}{N} \sum_{i=1}^{N} \log \frac{e^{sim(\mathbf{Z}_i^\alpha, \mathbf{Z}_i^\beta)/\tau}}{\sum_{j=1, j \neq i}^{N} e^{sim(\mathbf{Z}_i^\alpha, \mathbf{Z}_j^\alpha)/\tau}},\tag{25}$$

where $N$ denotes the batch size, $\tau$ is the temperature coefficient, and $sim(\cdot, \cdot)$ stands for cosine similarity function.

**Proposition C.4** (Lower Bound of $I(f(G^*); f(G))$).

$$I(f(G^*); f(G)) \geq -\mathcal{L}_{Cl}(G^*, G) + \log(N).\tag{26}$$

*Proof.* From [27], we easily get a lower bound of mutual information between $\mathbf{Z}_i^\alpha$ and $\mathbf{Z}_i^\beta$.

$$\mathbb{E}_{(G^\alpha, G^\beta)} \left[ \log \frac{e^{sim(\mathbf{Z}_i^\alpha, \mathbf{Z}_i^\beta)}}{\sum_{j=1, j \neq i}^{N} e^{sim(\mathbf{Z}_i^\alpha, \mathbf{Z}_j^\alpha)}} \right] = \frac{1}{N} \sum_{i=1}^{N} \log \frac{e^{sim(\mathbf{Z}_i^\alpha, \mathbf{Z}_i^\beta)}}{\sum_{j=1, j \neq i}^{N} e^{sim(\mathbf{Z}_i^\alpha, \mathbf{Z}_j^\alpha)}} \leq I(\mathbf{Z}_i^\alpha, \mathbf{Z}_i^\beta) - \log(N).\tag{27}$$

According to Eq. (25), we get

$$\mathcal{L}_{Cl}(G^\alpha, G^\beta) = -\mathbb{E}_{(G^\alpha, G^\beta)} \left[ \log \frac{e^{sim(\mathbf{Z}_i^\alpha, \mathbf{Z}_i^\beta)}}{\sum_{j=1, j\neq i}^N e^{sim(\mathbf{Z}_i^\alpha, \mathbf{Z}_j^\alpha)}} \right] = -\frac{1}{N} \sum_{i=1}^N \log \frac{e^{sim(\mathbf{Z}_i^\alpha, \mathbf{Z}_i^\beta)}}{\sum_{j=1, j\neq i}^N e^{sim(\mathbf{Z}_i^\alpha, \mathbf{Z}_j^\alpha)}}. \tag{28}$$

Thus, minimizing $\mathcal{L}_{Cl}(G^\alpha, G^\beta)$ is equivalent to maximize the lower bound of $I(\mathbf{Z}_i^\alpha, \mathbf{Z}_i^\beta)$.

Therefore, we have

$$-\mathcal{L}_{Cl}(G^\alpha, G^\beta) \leq I(\mathbf{Z}_i^\alpha, \mathbf{Z}_i^\beta) - \log(N), \tag{29}$$

which is equivalent to:

$$I(f(G^*); f(G)) \geq -\mathcal{L}_{Cl}(G^*, G) + \log(N). \tag{30}$$

This indicates that minimizing the contrastive loss $\mathcal{L}_{Cl}(G^*, G)$ is equivalent to maximizing the lower bound of the mutual information $I(f(G^*); f(G))$.

$\square$

## C.4 Proof for Proposition C.7

We first apply the upper bound proposed in the Variational Information Bottleneck [1].

**Lemma C.5** (Variational Upper Bound of Mutual Information). Given any two variables $X$ and $Y$, we have the variational upper bound of $I(X; Y)$:

$$\begin{aligned} I(X; Y) &= \mathbb{E}_{\mathbb{P}(X,Y)} \left[ \log \frac{\mathbb{P}(Y|X)}{\mathbb{P}(Y)} \right] = \mathbb{E}_{\mathbb{P}(X,Y)} \left[ \log \frac{\mathbb{P}(Y|X)\mathbb{Q}(Y)}{\mathbb{P}(Y)\mathbb{Q}(Y)} \right] \\ &= \mathbb{E}_{\mathbb{P}(X,Y)} \left[ \log \frac{\mathbb{P}(Y|X)}{\mathbb{Q}(Y)} \right] - \underbrace{D_{\mathrm{KL}} \left[ \mathbb{P}(Y) \| \mathbb{Q}(Y) \right]}_{\text{non-negative}} \\ &\leq \mathbb{E}_{\mathbb{P}(X,Y)} \left[ \log \frac{\mathbb{P}(Y|X)}{\mathbb{Q}(Y)} \right]. \end{aligned} \tag{31}$$

**Lemma C.6** (Variational Upper Bound of Conditional Mutual Information). For any $\mathbb{Q}(U|Y)$,

$$I(U; V|Y) \leq \mathbb{E}_{\mathbb{P}(U,V,Y)} \left[ \log \frac{\mathbb{P}(V|U,Y)}{\mathbb{Q}(V|Y)} \right]. \tag{32}$$

*Proof.* According to Lemma C.5, we can derive a variational upper bound for the conditional mutual information $I(U; V|Y)$. First, the conditional mutual information can be expressed as:

$$I(U; V|Y) = \mathbb{E}_{\mathbb{P}(U,V,Y)} \left[ \log \frac{\mathbb{P}(U, V|Y)}{\mathbb{P}(U|Y)\mathbb{P}(V|Y)} \right]. \tag{33}$$

Using the properties of conditional probability, we have

$$\mathbb{P}(U, V|Y) = \mathbb{P}(V|U, Y)\mathbb{P}(U|Y). \tag{34}$$

Substituting this into the expression for mutual information, we obtain:

$$I(U; V|Y) = \mathbb{E}_{\mathbb{P}(U,V,Y)} \left[ \log \frac{\mathbb{P}(V|U, Y)\mathbb{P}(U|Y)}{\mathbb{P}(U|Y)\mathbb{P}(V|Y)} \right] = \mathbb{E}_{\mathbb{P}(U,V,Y)} \left[ \log \frac{\mathbb{P}(V|U, Y)}{\mathbb{P}(V|Y)} \right]. \tag{35}$$

Next, introduce a variational distribution $\mathbb{Q}(V|Y)$ for $V$, and utilize the logarithmic identity:

$$\log \frac{\mathbb{P}(V|U, Y)}{\mathbb{P}(V|Y)} = \log \frac{\mathbb{P}(V|U, Y)}{\mathbb{Q}(V|Y)} - \log \frac{\mathbb{P}(V|Y)}{\mathbb{Q}(V|Y)}. \tag{36}$$

Therefore, the mutual information can be re-expressed as:

$$
\begin{aligned}
I(U;V|Y) &= \mathbb{E}_{\mathbb{P}(U,V,Y)}\left[\log\frac{\mathbb{P}(V|U,Y)}{\mathbb{Q}(V|Y)}\right] - \mathbb{E}_{\mathbb{P}(V,Y)}\left[\log\frac{\mathbb{P}(V|Y)}{\mathbb{Q}(V|Y)}\right] \\
&= \mathbb{E}_{\mathbb{P}(U,V,Y)}\left[\log\frac{\mathbb{P}(V|U,Y)}{\mathbb{Q}(V|Y)}\right] - \underbrace{D_{\mathrm{KL}}[\mathbb{P}(V|Y)\|\mathbb{Q}(V|Y)]}_{\text{non-negative}}.
\end{aligned}
\tag{37}
$$

Since the KL divergence is non-negative, we have a variational upper bound for the conditional mutual information:

$$
I(U;V|Y) \leq \mathbb{E}_{\mathbb{P}(U,V,Y)}\left[\log\frac{\mathbb{P}(V|U,Y)}{\mathbb{Q}(V|Y)}\right].
\tag{38}
$$

Similarly, if we introduce a variational distribution $\mathbb{Q}(U|Y)$ for $U$:

$$
I(U;V|Y) \leq \mathbb{E}_{\mathbb{P}(U,V,Y)}\left[\log\frac{\mathbb{P}(U|V,Y)}{\mathbb{Q}(U|Y)}\right].
\tag{39}
$$

In summary, the conditional mutual information $I(U;V|Y)$ can be upper bounded as:

$$
I(U;V|Y) \leq \mathbb{E}_{\mathbb{P}(U,V,Y)}\left[\log\frac{\mathbb{P}(V|U,Y)}{\mathbb{Q}(V|Y)}\right] \text{ or } I(U;V|Y) \leq \mathbb{E}_{\mathbb{P}(U,V,Y)}\left[\log\frac{\mathbb{P}(U|V,Y)}{\mathbb{Q}(U|Y)}\right].
\tag{40}
$$

$\square$

**Proposition C.7** (Upper Bound of $I(f(G^*);f(G)|\tilde{Y})$). For any $\mathbb{Q}(f(G)|\tilde{Y})$, the variational upper bound of conditional mutual information $I(f(G^*);f(G)|\tilde{Y})$ is:

$$
I(f(G^*);f(G)|\tilde{Y}) \leq \mathbb{E}_{\mathbb{P}}\left[\log\frac{\mathbb{P}(f(G),f(G^*)|\tilde{Y})}{\mathbb{P}(f(G^*)|\tilde{Y})\mathbb{Q}(f(G)|\tilde{Y})}\right].
\tag{41}
$$

*Proof.* Since $I(U;V|Y) \leq \mathbb{E}_{\mathbb{P}(U,V,Y)}\left[\log\frac{\mathbb{P}(V|U,Y)}{\mathbb{Q}(V|Y)}\right]$ in Lemma C.6, let $U = f(G^*)$, $V = f(G)$, we have

$$
I(f(G^*);f(G)|\tilde{Y}) \leq \mathbb{E}_{\mathbb{P}}\left[\log\frac{\mathbb{P}(f(G)|f(G^*),\tilde{Y})}{\mathbb{Q}(f(G)|\tilde{Y})}\right],
\tag{42}
$$

where $\mathbb{P}(f(G)|f(G^*),\tilde{Y})$ is the true conditional distribution $\mathbb{P}(f(G)|f(G^*),\tilde{Y})$, and $\mathbb{Q}(f(G)|\tilde{Y})$ is a variational distribution used to approximate $\mathbb{P}(f(G)|\tilde{Y})$. Based on the multiplication rule,

$$
\mathbb{P}(f(G),f(G^*)|\tilde{Y}) = \mathbb{P}(f(G)|f(G^*),\tilde{Y})\mathbb{P}(f(G^*)|\tilde{Y}).
\tag{43}
$$

This implies that

$$
\mathbb{P}(f(G)|f(G^*),\tilde{Y}) = \frac{\mathbb{P}(f(G),f(G^*)|\tilde{Y})}{\mathbb{P}(f(G^*)|\tilde{Y})}.
\tag{44}
$$

Substituting this into the variational upper bound, we have

$$
\begin{aligned}
I(f(G^*);f(G)|\tilde{Y}) &\leq \mathbb{E}_{\mathbb{P}}\left[\log\frac{\mathbb{P}(f(G)|f(G^*),\tilde{Y})}{\mathbb{Q}(f(G)|\tilde{Y})}\right] \\
&= \mathbb{E}_{\mathbb{P}}\left[\log\frac{\mathbb{P}(f(G),f(G^*)|\tilde{Y})}{\mathbb{P}(f(G^*)|\tilde{Y})\mathbb{Q}(f(G)|\tilde{Y})}\right].
\end{aligned}
\tag{45}
$$

$\square$

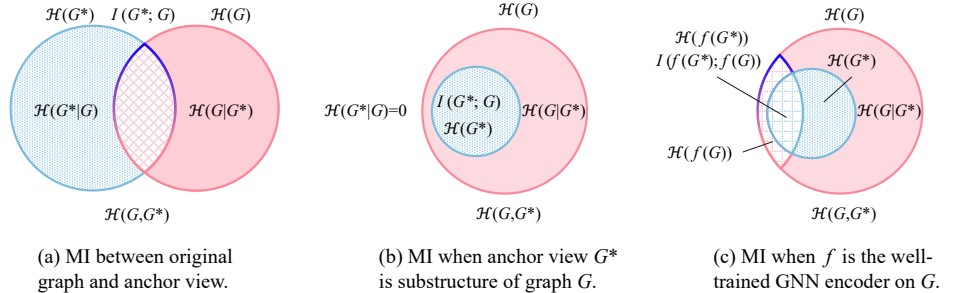

(a) MI between original graph and anchor view.

(b) MI when anchor view $G^*$ is substructure of graph $G$.

(c) MI when $f$ is the well-trained GNN encoder on $G$.

Figure 9: Venn diagram of the mutual information between original graph and essential view.

## C.5  Proof of Theorem C.9

In this section, we present the proof of the statement $\mathcal{H}(G^*|G) = 0$. First, we repeat the property that the target essential view should own:

**Definition C.8.** The essential view with redundancy-eliminated information is supposed to be a distinctive substructure of the given graph.

Now, let $G^*$ be the target essential view of graph $G$, the MI between $G$ and $G^*$ (*i.e.*, $I(G^*; G)$) can be formulated as:

$$I(G^*; G) = \mathcal{H}(G^*) - \mathcal{H}(G^*|G), \tag{46}$$

where $\mathcal{H}(G^*)$ is the structural entropy of $G^*$ and $\mathcal{H}(G^*|G)$ is the conditional entropy of $G^*$ conditioned on $G$. Before the proof, as shown in Figure 9, the Venn diagram on the left suggests the mutual information (MI) between the original graph and the essential view in general cases, while the Venn diagram on the right reveals the MI relationship when the essential view is a substructure of the given original graph (*i.e.*, complies with Definition C.8).

**Theorem C.9.** The information in $G^*$ is a subset of information in $G$ (*i.e.*, $\mathcal{H}(G^*) \subseteq \mathcal{H}(G)$); thus, we have

$$\mathcal{H}(G^*|G) = 0. \tag{47}$$

*Proof.* According to the definition of Shannon entropy [31], *i.e.*,

$$\mathcal{H}(X) = -\sum_{x \in X} \mathcal{P}(x) \log \mathcal{P}(x), \tag{48}$$

we follow the formulation of graph MI in [35] that $X$ is a set of node representations drawn from an empirical probability distribution of graph $G$, we have

$$\mathcal{H}(G^*|G) = \mathcal{H}(X^*|X). \tag{49}$$

So the conditional entropy can be written as:

$$\begin{aligned}
\mathcal{H}(X^*|X) &= \sum_{x \in X} \mathcal{P}(x)\mathcal{H}(X^*|X = x) \\
&= -\sum_{x \in X} \mathcal{P}(x) \sum_{x^* \in X^*} \mathcal{P}(x^*|x) \log \mathcal{P}(x^*|x) \\
&= -\sum_{x \in X} \sum_{x^* \in X^*} \mathcal{P}(x^*, x) \log \mathcal{P}(x^*|x) \\
&= -\sum_{x^*, x} \mathcal{P}(x^*, x) \log \mathcal{P}(x^*|x).
\end{aligned} \tag{50}$$

Considering that $G^*$ complies with the Definition C.8, the illustration of the probability distribution of $G^*$ and $G$ is shown Figure 9(b). Here, let us first discuss that when $x \in X$ and $x \notin X^*$, we have

$$\mathcal{P}(x^*, x) = 0. \tag{51}$$

Therefore, we can transform Eq. (50) to:

$$\mathcal{H}(X^*|X) = -\sum_{x^*,x} \mathcal{P}(x^*,x) \log \mathcal{P}(x^*|x)$$

$$= -\sum_{x^*,x^*} \mathcal{P}(x^*,x^*) \log \mathcal{P}(x^*|x^*) - \sum_{x^*,x \notin X^*} \mathcal{P}(x^*,x) \log \mathcal{P}(x^*|x)$$

$$= -\sum_{x^* \in X^*} \sum_{x^* \in X^*} \mathcal{P}(x^*,x^*) \log \mathcal{P}(x^*|x^*)$$

$$= -\sum_{x^* \in X^*} \mathcal{P}(x^*) \sum_{x^* \in X^*} \mathcal{P}(x^*|x^*) \log \mathcal{P}(x^*|x^*)$$

$$= \sum_{x^* \in X^*} \mathcal{P}(x^*) \mathcal{H}(X^*|X^* = x^*)$$

$$= \mathcal{H}(X^*|X^*)$$

$$= 0. \tag{52}$$

Therefore, given $\forall x \in X$, we have

$$\mathcal{H}(G^*|G) = \mathcal{H}(X^*|X) = 0. \tag{53}$$

Accordingly, we have

$$I(G^*; G) = \mathcal{H}(G^*). \tag{54}$$

$\square$

## C.6 Proof of Theorem C.12

With the essential view eliminating redundant information, we also theoretically prove that our RedOUT effectively captures the maximum mutual information between the representations obtained from the essential view and labels by minimizing the contrastive loss in Eq. (1). The InfoMax principle has been widely applied in representation learning literature [2, 28, 36]. MI quantifies the amount of information obtained about one random variable by observing the other random variable. We first introduce two lemmas.

**Lemma C.10.** Given that $f$ is a GNN encoder with learnable parameters and $G^*$ is the target essential view of graph $G$. If $I(f(G^*); f(G))$ reaches its maximum, then $I(f(G^*); G)$ will also reach its maximum.

*Proof.* Because $f(G)$ is a function of $G$,

$$I(f(G^*); G) = I(f(G^*); f(G); G) + I(f(G^*); G|f(G))$$
$$= I(f(G^*); f(G)) + I(f(G^*); G|f(G)). \tag{55}$$

Thus,

$$I(f(G^*); f(G)) = I(f(G^*); G) - I(f(G^*); G|f(G)). \tag{56}$$

While maximizing $I(f(G^*); f(G))$, either $I(f(G^*); G)$ increases or $I(f(G^*); G|f(G))$ decreases. When $I(f(G^*); G|f(G))$ reaches it minimum value of 0, $I(f(G^*); G)$ will definitely increase. Hence, the process of maximizing $I(f(G^*); f(G))$ can lead to the maximization of $I(f(G^*); G)$ as well. $\square$

**Lemma C.11.** Given the essential view $G^*$ of graph $G$ and an encoder $f$, we have

$$I(f(G^*); G) \leq I(f(G^*); Y) + I(G; G^*|Y). \tag{57}$$

*Proof.*

$$I(f(G^*); G) = I(f(G^*); G; Y) + I(f(G^*); G|Y)$$
$$= I(f(G^*); Y) - I(f(G^*); Y|G) + I(f(G^*); G|Y). \tag{58}$$

Due to the non-negativity of mutual information,

$$I(G; G^*|Y) = I(f(G^*); G; G^*|Y) + I(G; G^*|Y, f(G^*))$$
$$\geq I(f(G^*); G; G^*|Y) \quad \text{(the non-negativity of } I\text{)} \tag{59}$$
$$= I(f(G^*); G|Y).$$

According to Eq. (58) and (59), we get

$$I(f(G^*); G) + I(f(G^*); Y|G) = I(f(G^*); Y + I(f(G^*); G|Y)$$
$$\leq I(f(G^*); Y) + I(G; G^*|Y). \tag{60}$$

Thus,

$$I(f(G^*); G) \leq I(f(G^*); Y) + I(G; G^*|Y). \tag{61}$$

$\square$

**Theorem C.12.** According to Proposition 2, optimizing the contrastive loss $\mathcal{L}_{Cl}(G^*, G)$ is equivalent to maximizing $I(f(G^*); f(G))$, which could lead to the maximization of $I(f(G^*); Y)$.

*Proof.* Minimizing the contrastive loss $\mathcal{L}_{Cl}(G^*, G)$ is equivalent to maximizing a lower bound of the mutual information between the latent representations of two views of the graph, and can be viewed as one way of mutual information maximization between the latent representations (*i.e.*, $\max I(f(G^*); f(G))$). Consequently, the optimization of $\mathcal{L}_{Cl}(G^*, G)$ is equivalent to maximizing $I(f(G^*); f(G))$.

According to Lemma C.10, we know that maximizing $I(f(G^*); f(G))$ is equivalent to maximizing $I(f(G^*); G)$, *i.e.*,

$$\max I(f(G^*); f(G)) \Rightarrow \max I(f(G^*); G). \tag{62}$$

According to Lemma C.11, we have

$$I(f(G^*); G) \leq I(f(G^*); Y) + I(G; G^*|Y). \tag{63}$$

Since

$$I(G; G^*) = I(G; G^*|Y) + I(G; G^*; Y), \tag{64}$$

it follows that

$$I(G; G^*|Y) \leq I(G; G^*). \tag{65}$$

By combining Eq. (63) and Eq. (65), we obtain:

$$I(f(G^*); G) \leq I(f(G^*); Y) + I(G; G^*|Y)$$
$$\leq I(f(G^*); Y) + I(G; G^*) \tag{66}$$

Thus,

$$I(f(G^*); G) - I(G; G^*) \leq I(f(G^*); Y). \tag{67}$$

Since the Eq. (54) in Theorem C.9 we have already proven that $I(G^*; G) = \mathcal{H}(G^*)$, and $G^*$ is the essential view of graph $G$ obtained by minimizing structural entropy (*i.e.*, $\min \mathcal{H}(G^*)$), it follows that

$$\min I(G; G^*) = \min \mathcal{H}(G^*). \tag{68}$$

Thus,

$$\max I(f(G^*); G) - I(G^*; G) \leq \max I(f(G^*); Y). \tag{69}$$

Therefore, minimizing the contrastive loss $\mathcal{L}_{Cl}(G^*, G)$ is equivalent to maximizing $I(f(G^*); f(G))$. At this point, $I(f(G^*); G)$ reaches its maximum, and $I(G^*; G)$ reaches its minimum, thereby maximizing $I(f(G^*); Y)$.

$\square$

## C.7 Proof of Theorem C.15

**Definition C.13** (Graph Quotient Space). Define the equivalence $\cong$ between two graphs $G_1 \cong G_2$ if $G_1, G_2$ cannot be distinguished by the 1-WL test. Define the quotient space $\mathcal{G}' = \mathcal{G}/\cong$.

So every element in the quotient space, *i.e.*, $G' \in \mathcal{G}'$, is a representative graph from a family of graphs that cannot be distinguished by the 1-WL test. Note that our definition also allows attributed graphs.

**Definition C.14** (Probability Measures in $\mathcal{G}'$). Define $\mathbb{P}_{\mathcal{G}'}$ over the space $\mathcal{G}'$ such that $\mathbb{P}_{\mathcal{G}'}(G') = \mathbb{P}_{\mathcal{G}}(G \cong G')$ for any $G' \in \mathcal{G}'$. Further define $\mathbb{P}_{\mathcal{G}' \times \mathcal{Y}}(G', Y') = \mathbb{P}_{\mathcal{G} \times \mathcal{Y}}(G \cong G', Y = Y')$. Given a GDA $T(\cdot)$ defined over $\mathcal{G}$, define a distribution on $\mathcal{G}'$, $\mathcal{T}'(G') = \mathbb{E}_{G \sim \mathbb{P}_{\mathcal{G}}}[\mathcal{T}(G)|G \cong G']$ for $G' \in \mathcal{G}'$.

**Theorem C.15** (Bounds of Redundant and Essential Information). Suppose the encoder $f$ is implemented by a GNN as powerful as the 1-WL test. Suppose $\mathcal{G}$ is a countable space and thus $\mathcal{G}'$ is a countable space, where $\cong$ denotes equivalence ($G_1 \cong G_2$ if $G_1, G_2$ cannot be distinguished by the 1-WL test). Define $\mathbb{P}_{\mathcal{G}' \times \mathcal{Y}}(G', Y') = \mathbb{P}_{\mathcal{G} \times \mathcal{Y}}(G \cong G', Y)$ and $\mathcal{T}'(G') = \mathbb{E}_{G \sim \mathbb{P}_G}[\mathcal{T}^*(G)|G \cong G']$ for $G' \in \mathcal{G}$. Then, the optimal $f^*$ and $G^*$ to RedOUT satisfies:

1. $I(G^*; Y) = I(f^*(G^*); Y) \geq I(t'(G'); Y)$,

2. $I(f^*(G); G|Y) \leq \min_{\mathcal{T}} I(t'(G'); G')) - I(t'(G'); Y)$,

where $t^*(G) = G^*, t'(G') \sim \mathcal{T}'(G'), t'^*(G') \sim \mathcal{T}'^*(G'), (G, Y) \sim \mathbb{P}_{\mathcal{G} \times \mathcal{Y}}$ and $(G', Y) \sim \mathbb{P}_{\mathcal{G}' \times \mathcal{Y}}$.

*Proof.* Because $\mathcal{G}$ and $\mathcal{G}'$ are countable, $\mathbb{P}_\mathcal{G}$ and $\mathbb{P}_{\mathcal{G}'}$ are defined over countable sets and thus discrete distribution. Later we may call a function $z(\cdot)$ can distinguish two graphs $G_1, G_2$ if $z(G_1) \neq z(G_2)$.

Moreover, for notational simplicity, we consider the following definition. Because $f^*$ is as powerful as the 1-WL test. Then, for any two graphs $G_1, G_2 \in \mathcal{G}$, $G_1 \cong G_2$, $f^*(G_1) = f^*(G_2)$. We may define a mapping over $\mathcal{G}'$, also denoted by $f^*$ which simply satisfies $f^*(G') :\triangleq f^*(G)$, where $G \cong G'$, and $G \in \mathcal{G}$ and $G' \in \mathcal{G}'$.

We first prove the statement 1, *i.e.*, the lower bound. Given $G^*$, $G^* \Rightarrow f^*(G^*)$ is an injective deterministic mapping because of the injective $f^*$. Therefore, we have

$$I(f^*(G); Y) = I(G^*; Y). \tag{70}$$

Given $t'^*(G')$, $t'^*(G') \to f^*(t'^*(G'))$ is an injective deterministic mapping. Therefore, for any random variable $Q$,

$$I(f^*(t'^*(G')); Q) = I(t'^*(G'); Q),$$

where $G' \sim \mathbb{P}_{\mathcal{G}'}, t'^*(G') \sim \mathcal{T}'^*(G')$.

Of course, we may set $Q = Y$. So,

$$I(f^*(t'^*(G')); Y) = I(t'^*(G'); Y), \tag{71}$$

where $(G', Y) \sim \mathbb{P}_{\mathcal{G}' \times \mathcal{Y}}, t'^*(G') \sim \mathcal{T}'^*(G')$.

Because of the data processing inequality [6] and $\mathcal{T}'^*(G') = \mathbb{E}_{G \sim \mathbb{P}_G}[\mathcal{T}^*(G)|G \cong G']$, we further have

$$I(f^*(t^*(G)); Y) \geq I(f^*(t'^*(G')); Y), \tag{72}$$

where $(G', Y) \sim \mathbb{P}_{\mathcal{G}' \times \mathcal{Y}}, (G, Y) \sim \mathbb{P}_{\mathcal{G} \times \mathcal{Y}}, t'^*(G') \sim \mathcal{T}'^*(G'), t^*(G) \sim \mathcal{T}^*(G)$. Further because of the data processing inequality [6],

$$I(f^*(G); Y) \geq I(f^*(t^*(G)); Y). \tag{73}$$

Combining Eq. (71), (72), (73), we have

$$I(f^*(G^*); Y) = I(f^*(t^*(G)); Y) \geq I(f^*(t'^*(G')); Y) = I(t'^*(G'); Y) \geq \min_{\mathcal{T}} I(t'(G'); Y), \tag{74}$$

which concludes the proof of the lower bound.

We next prove the statement 2, *i.e.*, the upper bound. Recall that $\mathcal{T}'^*(G') = \mathbb{E}_{G \sim \mathbb{P}_G}[\mathcal{T}^*(G)|G \cong G']$ and $t'^*(G') \sim \mathcal{T}'^*(G')$.

$$
\begin{aligned}
I(t'^*(G'); G') &= I(t'^*(G'); (G', Y)) - I(t'^*(G'); Y|G')] \\
&\overset{(a)}{=} I(t'^*(G'); (G', Y)) \\
&= I(t'^*(G'); Y) + I(t'^*(G'); G'|Y) \\
&\overset{(b)}{\geq} I(f^*(t'^*(G')); G'|Y) + I(t'^*(G'); Y)
\end{aligned}
\tag{75}
$$

where $(a)$ is because $t'^*(G') \perp_{G'} Y$, $(b)$ is because the data processing inequality [6]. Moreover, because $f^*$ could be as powerful as the 1-WL test and thus could be injective in $\mathcal{G}'$ a.e. with respect to the measure $\mathbb{P}_{\mathcal{G}'}$. Since $f^*(G) = f^*(G')$ and $\mathcal{T}'(G') = \mathbb{E}_{G \sim \mathbb{P}_{\mathcal{G}}}[\mathcal{T}(G)|G \cong G']$,

$$I(t'(G'); G') = I(f^*(t'(G')); f^*(G')) = I(f^*(t(G)); f^*(G)),\tag{76}$$

where $t'(G') \sim \mathcal{T}'(G')$, $t(G) \sim \mathcal{T}(G)$.

Since $\mathcal{T}^* = \arg\min_{\mathcal{T}} I(G^*, G)$ where $t^*(G) \sim \mathcal{T}^*(G)$ and Eq. (76), we have

$$I(t'^*(G'); G') = \arg\min_{\mathcal{T}} I(t'(G'); G'),\tag{77}$$

Again, because $f^*$ could be as powerful as the 1-WL test, its counterpart defined over $\mathcal{G}'$, *i.e.*, $f^\star$ must be injective over $\mathcal{G}' \cap \mathrm{Supp}(\mathbb{E}_{G' \sim \mathbb{P}_{\mathcal{G}'}}[\mathcal{T}'^*(G')])$ a.e. with respect to the measure $\mathbb{P}_{\mathcal{G}'}$ to achieve such mutual information maximization. Here, $\mathrm{Supp}(\mu)$ defines the set where $\mu$ has non-zero measure.

Because of the definition of $\mathcal{T}'^*(G') = \mathbb{E}_{G \sim \mathbb{P}_{\mathcal{G}}}[\mathcal{T}^*(G)|G \cong G']$,

$$\mathcal{G}' \cap \mathrm{Supp}(\mathbb{E}_{G' \sim \mathbb{P}_{\mathcal{G}'}}[\mathcal{T}'^*(G')]) = \mathcal{G}' \cap \mathrm{Supp}(\mathbb{E}_{G \sim \mathbb{P}_{\mathcal{G}}}[\mathcal{T}^*(G)]).\tag{78}$$

Therefore, $f^*$ is a.e. injective over $\mathcal{G}' \cap \mathrm{Supp}(\mathbb{E}_{G \sim \mathbb{P}_{\mathcal{G}}}[\mathcal{T}^*(G)])$ and thus

$$I(f^*(t'^*(G')); G'|Y) = I(f^*(t^*(G)); G'|Y),\tag{79}$$

Moreover, as $f^*$ cannot cannot distinguish more graphs in $\mathcal{G}$ than $\mathcal{G}'$ as the power of $f^*$ is limited by 1-WL test, thus,

$$I(f^*(t^*(G)); G'|Y) = I(f^*(t^*(G)); G|Y).\tag{80}$$

Plugging Eq. (77),(79),(80) into Eq. (75), we achieve

$$I(f^*(G^*); G|Y) = I(f^*(t^*(G)); G|Y) \le \arg\min_{\mathcal{T}} I(t'(G'); G') - I(t'^*(G'); Y)$$

$$\le \arg\min_{\mathcal{T}} I(t'(G'); G') - I(t'(G'); Y).\tag{81}$$

where $t'(G') \sim \mathcal{T}'(G') = \mathbb{E}_{G \sim \mathbb{P}_{\mathcal{G}}}[\mathcal{T}(G)|G \cong G']$ and $t'^*(G') \sim \mathcal{T}'^*(G') = \mathbb{E}_{G \sim \mathbb{P}_{\mathcal{G}}}[\mathcal{T}^*(G)|G \cong G']$, which gives us the statement 1, which is the upper bound.

$\square$

i) Enhancing Essential Information: statement 1 of Theorem C.15 highlights the effectiveness of ReGIB in capturing essential information. It implies that the essential information $G^*$ is guaranteed to have at least as much mutual information with the ground-truth labels $Y$ as the augmented representation $t'(G')$, which suggests that $G^*$ is highly informative with respect to the downstream task.

ii) Limiting Redundant Information: statement 2 of Theorem C.15 establishes an upper bound on the redundant information embedded in the representations. This aligns with the GIB principle (Eq. (3) when $\beta = 1$), ensuring that the encoder $f^*$ captures only the necessary information from input graph $G$ that is relevant to the downstream task. The statement suggests that ReGIB is capable of producing representations with limited redundant information, thereby enhancing the overall efficiency of the representation learning process.

## C.8 Proof of Instantiation for $I(f(G^*); f(G)|\tilde{Y})$

According to Proposition C.7, we have:

$$I(f(G^*); f(G)|\tilde{Y}) \le \mathbb{E}_{\mathbb{P}}\left[\log \frac{\mathbb{P}(f(G), f(G^*)|\tilde{Y})}{\mathbb{P}(f(G^*)|\tilde{Y})\mathbb{Q}(f(G)|\tilde{Y})}\right].\tag{82}$$

To specify the variational upper bound, we provide the proof for the instantiation of $I(f(G^*); f(G)|\tilde{Y})$ as follows.

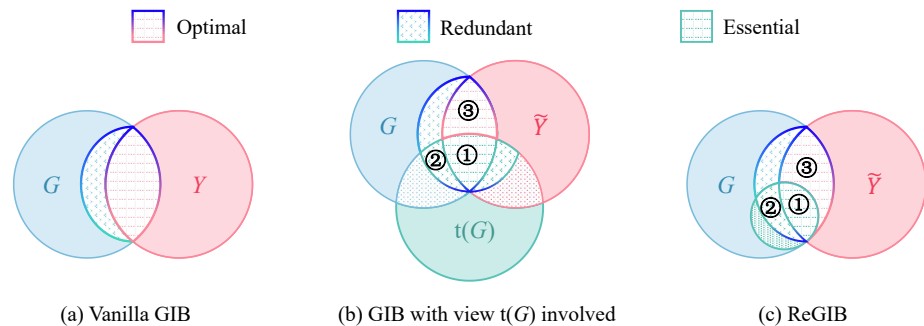

(a) Vanilla GIB        (b) GIB with view t($G$) involved        (c) ReGIB

Figure 10: Venn diagram of comparison between the basic GIB, GIB with a random augmentation view, and proposed ReGIB. Note ReGIB captures the essential information and discards redundancy, wherein (c) ①+③ = $I(f(G); \tilde{Y})$, ①+② = $I(f(G^*); f(G))$, ① = $I(f(G^*); \tilde{Y})$.

*Proof.* Since

$$
\begin{aligned}
I(U; V|Y) &\le \mathbb{E}_{P(U,V,Y)} \left[ \log \frac{e^{sim(u,v)}}{\frac{1}{N}\sum_{i=1}^{N} e^{sim(u,v_i^-)}} \right] \\
&= \mathbb{E}_{P(U,V,Y)} \left[ sim(u,v) - \log\left(\frac{1}{N}\sum_{i=1}^{N} e^{sim(u,v_i^-)}\right) \right].
\end{aligned}
\tag{83}
$$

Here, we give the definition of the conditional redundancy-eliminated loss $\mathcal{L}_{CRI}$:

$$
\mathcal{L}_{CRI}(G, G^*) \triangleq \mathbb{E}_{\tilde{y}\sim\mathbb{P}(Y)} \mathbb{E}_{\mathbf{Z},\mathbf{Z}_T \sim \mathbb{P}(f(G), f(G^*)|\tilde{y})} \left[ sim(\mathbf{Z}, \mathbf{Z}_T) - \log \mathbb{E}_{\mathbf{Z}^-\sim\mathbb{P}(f(G)|\tilde{y})} e^{sim(\mathbf{Z}^-, \mathbf{Z}_T)} \right].
\tag{84}
$$

Now, for Eq. (82), we treat the similarity between $\mathbf{Z}_T$ and $\mathbf{Z}$ given the predicted predicted label $\tilde{Y} = \text{softmax}(\mathbf{Z})$ as an approximation of the log-likelihood $\log\mathbb{P}(f(G)|f(G^*), \tilde{Y})$, and set

$$
\mathbb{Q}(f(G)|\tilde{Y}) = \mathbb{E}_{\mathbf{Z}^-\sim\mathbb{P}(f(G)|\tilde{Y})} e^{\text{sim}(\mathbf{Z}^-, \mathbf{Z}_T)},
\tag{85}
$$

where $\mathbf{Z}^-$ are negative samples drawn from the conditional distribution $\mathbb{P}(f(G)|\tilde{Y})$.

Thus, $I(f(G^*); f(G) \mid \tilde{Y})$ can be approximately instantiated as:

$$
I(f(G^*); f(G)|\tilde{Y}) \doteq \mathcal{L}_{CRI}(G, G^*),
\tag{86}
$$

$\square$

## C.9    Further Comparison of ReGIB and GIB with Random Augmentation Views

In this section, we provide a comparative analysis of ReGIB and GIB with random augmentation views from an argumentation perspective. Compared to the standard GIB with random augmentation views ($\triangleright$ Figure 10(a)), ReGIB addresses the limitations of models pretrained solely on ID data. Such models often lack the generalization capability required for OOD data, which results in representations that contain both relevant and irrelevant information ($\triangleright$ Figure 10(c)). For unsupervised tasks, models pretrained on ID data can leverage pseudo-labels $\tilde{Y}$ obtained from softmax outputs to reduce dependence on ground-truth labels $Y$.

As discussed previously, ID and OOD graphs can be distinguished by their unique structural characteristics. Through the proof of Theorem C.9, we show that essential structural information is embedded within the original graph and can be extracted by minimizing structural entropy to obtain the essential view $G^*$. Methods such as GOODAT [39], however, which use graph maskers to identify subgraphs in test data similar to those in ID graphs, fundamentally rely on learnable graph augmentations. These augmentations may inadvertently alter the semantic information of substructures or lead to information loss, compromising the reliability of OOD detection. Specifically, as shown in Figure 10(b)) random augmentations on graph $G$ can introduce redundant information captured by the pretrained model, making it more difficult to extract the optimal and essential information.

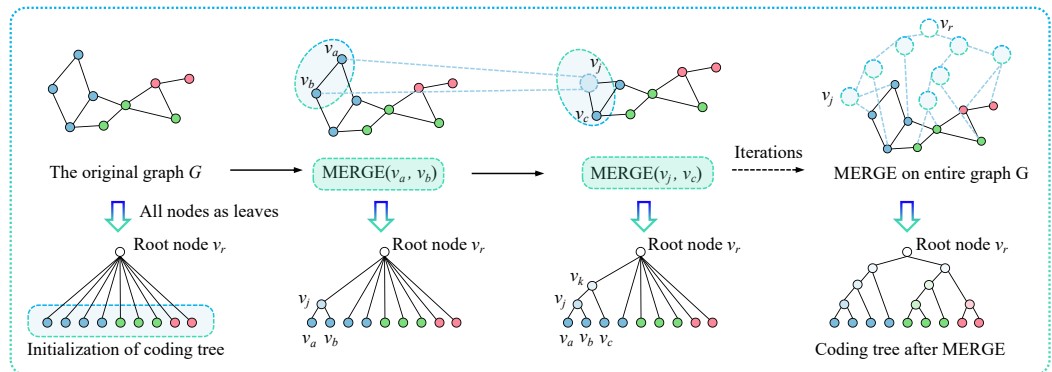

Figure 11: Overview of **MERGE** to construct full-height binary coding tree.

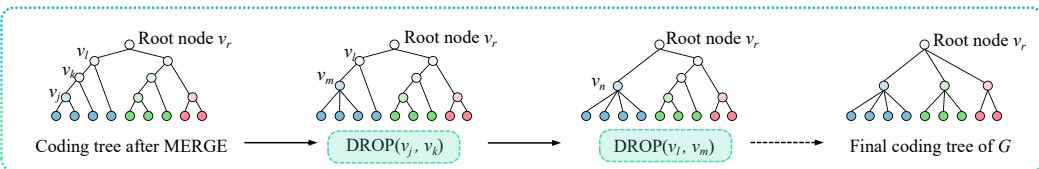

Figure 12: Overview of **DROP** to squeeze the height of coding tree to $k$.

Since well-trained GNNs generally perform well on ID graphs but tend to produce more random and unreliable predictions for unseen OOD graphs in the test data, their predictions fail to identify distinctive structural information or recognize similar modifying structures. For graph data with specific structures, the intrinsic semantic information is encoded in the structure itself. Unlike $t(G)$ which introduces random irrelevant information, $G^*$ is a hierarchical abstraction derived from the original graph structure, obtained by minimizing the structural entropy of $G$. This process preserves the essential structure without altering the semantic information. Therefore, the distinctive pattern of $G^*$ serves as a form of ground-truth information to correct predictions on test graphs.

The primary objective of ReGIB is to obtain the distinctive structure $G^*$ and to extract as much optimal and essential information as possible based on pseudo-labels while eliminating irrelevant redundancies. Thus, ReGIB can be considered a special case of GIB with random augmentation views. By placing $t(G)$ entirely within the space of $G$ in vanilla GIB, we derive ReGIB. In summary, ReGIB eliminates irrelevant redundancies introduced by random modifications, effectively simplifying the representation of the graph's intrinsic structure and extracting optimal essential information without adding unnecessary redundancies. In unlabeled OOD detection tasks and test-time settings that rely solely on test samples, ReGIB proves to be more effective.

## D   Algorithms

To start with, we initialize a coding tree $T$ by treating all nodes in $\mathcal{V}$ as children of root node $v_r$. The construction of the coding tree involves initially building a full-height binary tree with all nodes as leaves and then optimizing this binary tree into a fixed-height coding tree. Specifically, during step 1, an iterative **MERGE**$(v_c^1, v_c^2)$ is performed with the goal of minimizing structural entropy to obtain a binary coding tree without height limitation. In this way, selected leaf nodes are combined to form new community divisions with minimal structural entropy. Then, to compress height to a specific hyper-parameter $k$, **DROP**$(v_m)$ is leveraged to merge small divisions into larger ones, and thus the height of the coding tree is reduced, which is still following the structural entropy minimization strategy. Eventually, the coding tree with fixed height $k$ and minimal structural entropy is obtained.

**Definition D.1.** Assuming $v_a$ and $v_b$ as two child nodes of root node $v_r$, the function **MERGE**$(v_a, v_b)$ is defined as adding a new node $v_j$ as the child of $v_r$ and the parent of $v_a$ and $v_b$:

$$
\begin{aligned}
v_j.children &= \{v_a, v_b\}, \\
v_r.children &= \{v_j\} \cup v_r.children.
\end{aligned} \tag{87}
$$

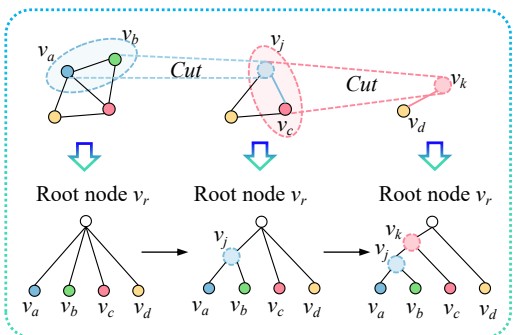

Figure 13: The process of adding nodes and cutting edges to the corresponding coding tree.

Since merging nodes in the original graph via the operator $\textbf{MERGE}(v_a, v_b)$ operator reduces the structural entropy of graph $G$, the pair of child nodes to be merged should be the one that maximizes the reduction in structural entropy, formally written as:

$$(v_a, v_b) = argmax\{\mathcal{H}^T(G) - \mathcal{H}^{T_{ab}}(G)|v_a, v_b \in v_r.children\}. \tag{88}$$

An overview of the $\textbf{MERGE}$ operator is shown in Figure 11. To provide a more detailed explanation of the coding tree construction process, we first revisit the definition of structural entropy as follows:

$$\mathcal{H}^T(G) = -\sum_{v_\tau \in T} \frac{g_{v_\tau}}{vol(\mathcal{V})} \log \frac{vol(v_\tau)}{vol(v_\tau^+)}, \tag{89}$$

where $v_\tau$ is a node in $T$ except for root node and also stands for a subset $\mathcal{V}_\tau \in \mathcal{V}$, $g_{v_\tau}$ is the number of edges connecting nodes in and outside $\mathcal{V}_\tau$, $v_\tau^+$ is the immediate predecessor of $v_\tau$ and $vol(v_\tau)$, $vol(v_\tau^+)$ and $vol(\mathcal{V})$ are the sum of degrees of nodes in $v_\tau$, $v_\tau^+$ and $\mathcal{V}$, respectively. When two nodes are merged into a new node (e.g., merging the pair $(v_a, v_b)$ into $v_j$), the structural information of the new node must satisfy the following equations:

$$\begin{aligned} g_{v_j} &= g_{v_a} + g_{v_b} - 2Cut(v_a, v_b), \\ vol(v_j) &= vol(v_a) + vol(v_b), \end{aligned} \tag{90}$$

where $Cut(v_a, v_b)$ denotes the number of edges cut between nodes when merging $(v_a, v_b)$. Figure 13 illustrates the process of merging nodes in a graph and adding nodes to the corresponding coding tree through an example.

**Definition D.2.** Given node $v_m$ and its parent node $v_m^+$ in $T$, the operator $\textbf{DROP}(v_m)$ is defined as adding the children of $v_m$ and itself to the child set of $v_m^+$:

$$v_m^+.children = v_m^+.children \cup v_m.children. \tag{91}$$

Similarly, since creating a new node $v_m$ through the $\textbf{DROP}(v_m)$ operator also changes the structural entropy of graph $G$, the selection of the new node $v_m$ should aim to minimize the change in structural entropy, written as:

$$v_m = argmin\{\mathcal{H}^{T_m}(G) - \mathcal{H}^T(G)|v_m \in T, v_m \neq v_r, v_m \notin \mathcal{V}\} \tag{92}$$

An overview of the $\textbf{DROP}$ operator is shown in Figure 12. The construction of coding tree with a fixed height $k$ primarily involves iterations through the two operators to obtain the minimum structural entropy, which is shown in Algorithm 1.

# E  Experiment

## E.1  Dataset Description

For OOD detection, we employ 10 pairs of datasets from two mainstream graph data benchmarks (i.e., TUDataset [24] and OGB [11]) following GOOD-D [20]. Specifically, we select 8 pairs of molecular

---

**Algorithm 1:** Coding tree construction with height $k$ via structural entropy minimization.

---

**Input:** A undirected graph $G = (\mathcal{V}, \mathcal{E})$; Specific height $k > 1$.
**Output:** Coding tree $T$ with height $k$.

1   Initialize a coding tree $T$ with root node $v_r$ and nodes in $\mathcal{V}$ as its children ;
2   // *Step 1: full-height binary coding tree construction*
3   **while** $|v_j.children| > 2$ **do**
4      Select child node pair $(v_a, v_b) \leftarrow$ Eq. (88) ;
5      **MERGE**$(v_a, v_b)$;
6   **end**
7   // *Step 2: binary coding tree squeeze to height $k$*
8   **while** $Height(T) > k$ **do**
9      Select node $v_m \leftarrow$ Eq. (92) ;
10     **DROP**$(v_m)$ ;
11   **end**
12   **return** coding tree $T$ ;

---

---

**Algorithm 2:** Overall optimization process of RedOUT.

---

**Input:** Test graph sample $G$; Pre-trained GNN encoder $f$ which is frozen; Coding tree encoder $f_\Theta$; Number of test-time training epochs $E$; Hyperparameters $\lambda$.
**Output:** Optimized tree encoder $f_\Theta^\star$; Predicted OOD score $S$.

1   Initialize parameters randomly;
2   // *Instantiation for Redundancy-eliminated Essential Information*
3   Construct redundancy-eliminated $G^*$ via structural entropy minimization $\leftarrow$ Eq. (14);
4   **for** $i = 1, 2, \cdots, E$ **do**
5      Obtain representations for original graph and coding tree, as $\mathbf{Z} = f(G)$, $\mathbf{Z}_T = f_\Theta(G^*)$;
6      // *Instantiation for Lower Bound of $I(f(G^*); f(G)$*
7      Calculate contrastive loss, as $\mathcal{L}_{Cl} = -I(\mathbf{Z}, \mathbf{Z}_T) \leftarrow$ Eq. (15);
8      // *Instantiation for Upper Bound of $I(f(G^*); f(G)|\tilde{Y})$*
9      Calculate conditional redundancy-eliminated loss, as $\mathcal{L}_{CRI} = I(\mathbf{Z}, \mathbf{Z}_T|\tilde{Y}) \leftarrow$ Eq. (16);
10     Calculate the overall loss, as $\mathcal{L} \leftarrow$ Eq. (18);
11     Update parameter $\Theta$ by minimizing $\mathcal{L}$ and back-propagation;
12   **end**

---

datasets, 1 pair of protein datasets, and 1 pair of social network datasets. $90\%$ of ID samples are used for training, and $10\%$ of ID samples and the same number of OOD samples are integrated together for testing. The partitioning of ID samples for training, along with the division of ID and OOD samples for testing, follows GOOD-D [20]. Detailed statistics of OOD detection datasets are compiled in Table 5. Further detailed information about these datasets is categorized and described as follows.

### E.1.1   Molecular Datasets

- **BZR** [24] is a dataset focused on benzodiazepine receptor ligands, containing molecular structures and associated binding affinities. It is crucial for drug design and discovery, specifically for studying receptor-ligand interactions.

- **PTC-MR** [24] reports the carcinogenicity of 344 chemical compounds in male and female rats and includes 19 discrete labels. It is utilized for predicting the carcinogenic potential of chemical substances.

- **AIDS** [24] contains data on anti-HIV compounds, including their molecular structures and biological activities, serving as a valuable resource for the development of anti-HIV drugs.

- **ENZYMES** [24] is a dataset consisting of protein structures classified into enzyme types based on their functionality. It is used for protein function prediction and enzyme classification.

- **COX2** [24] comprises data on cyclooxygenase-2 inhibitors, which are compounds with anti-inflammatory properties. This dataset is essential for the research and development of anti-inflammatory drugs.

Table 5: Statistics of OOD detection datasets.

| Dataset Pair | Domain | #ID train | #ID test | #OOD test |
|---|---|---|---|---|
| BZR / COX2 | Molecules | 364 | 41 | 41 |
| PTC-MR / MUTAG | Molecules | 309 | 35 | 35 |
| AIDS / DHFR | Molecules | 1800 | 200 | 200 |
| Tox21 / SIDER | Molecules | 7047 | 784 | 784 |
| FreeSolv / ToxCast | Molecules | 577 | 65 | 65 |
| BBBP / BACE | Molecules | 1835 | 204 | 204 |
| ClinTox / LIPO | Molecules | 1329 | 148 | 148 |
| Esol / MUV | Molecules | 1015 | 113 | 113 |
| ENZYMES / PROTEINS | Proteins | 540 | 60 | 60 |
| IMDB-M / IMDB-B | Social Networks | 1350 | 150 | 150 |

Table 6: Further statistics of graph datasets.

| Dataset | #Feature | #Graphs | Avg. Nodes | Avg. Edges | Avg. Deg. |
|---|---|---|---|---|---|
| ENZYMES | 1 | 600 | 32.63 | 62.13 | 1.90 |
| PROTEIN | 1 | 1113 | 39.06 | 72.82 | 1.86 |
| IMDB-M | 1 | 1500 | 18.00 | 65.93 | 5.07 |
| IMDB-B | 1 | 1000 | 19.77 | 96.53 | 4.88 |
| Tox21 | 9 | 7831 | 18.57 | 19.29 | 1.04 |
| SIDER | 9 | 1427 | 33.64 | 35.66 | 1.05 |
| FreeSolv | 9 | 642 | 8.72 | 8.38 | 0.96 |
| ToxCast | 9 | 8576 | 18.78 | 19.26 | 1.03 |
| BBBP | 9 | 2039 | 24.06 | 25.95 | 1.08 |
| BACE | 9 | 1513 | 34.08 | 35.91 | 1.08 |
| ClinTox | 9 | 1477 | 26.15 | 27.88 | 1.07 |
| LIPO | 9 | 4200 | 27.04 | 29.11 | 1.09 |
| Esol | 9 | 1128 | 13.28 | 14.08 | 1.03 |
| MUV | 9 | 93087 | 24.23 | 26.27 | 1.08 |
| BZR | 1 | 405 | 35.75 | 38.14 | 1.07 |
| COX2 | 1 | 467 | 41.22 | 44.52 | 1.05 |
| PTC_MR | 1 | 344 | 14.28 | 15.04 | 1.03 |
| MUTAG | 1 | 188 | 17.93 | 19.79 | 1.10 |

- **MUTAG** [24] has seven kinds of graphs derived from 188 mutagenic aromatic and heteroaromatic nitro compounds. It is used for studying the mutagenicity of chemical substances.

- **DHFR** [24] includes dihydrofolate reductase inhibitors, important in the development of antibacterial and anticancer drugs, aiding in drug discovery and medicinal chemistry research.

- **PROTEINS** [24] contains data on protein structures and their functionalities. Nodes represent secondary structure elements (SSEs), and edges connect neighboring elements in the amino acid sequence or 3D space. This dataset is used for protein structure prediction and functional analysis.

- **Tox21** [11] is a dataset containing toxicity data on 12 biological targets, which has been used in the 2014 Tox21 Data Challenge and includes nuclear receptors and stress response pathways.

- **BBBP** [11, 23] includes records of whether a compound has the permeability property of penetrating the blood-brain barrier, essential for the design of central nervous system drugs.

- **ClinTox** [11, 26, 8] contains clinical toxicity data on a variety of drug compounds, classifying drugs approved by the FDA and those that have failed clinical trials for toxicity reasons.

- **ToxCast** [11, 29] includes high-throughput screening data on the toxicity of chemical substances, with measurements based on over 600 in vitro screenings. This dataset is used for large-scale toxicity assessment and environmental health research.

- **SIDER** [11, 14] contains information on drug side effects, grouped into 27 system organ classes, also known as the Side Effect Resource. It is utilized for predicting drug side effects and improving drug safety profiles.

Table 7: Statistics of anomaly detection datasets.

| Dataset Pair | Domain | #ID train | #ID test | #OOD test |
|---|---|---|---|---|
| BZR | Molecules | 69 | 17 | 64 |
| AIDS | Molecules | 1280 | 320 | 80 |
| COX2 | Molecules | 81 | 21 | 73 |
| NCI1 | Molecules | 1646 | 411 | 411 |
| DHFR | Molecules | 368 | 93 | 59 |
| ENZYMES | Proteins | 400 | 100 | 20 |
| PROTEINS | Proteins | 360 | 90 | 133 |
| DD | Proteins | 390 | 97 | 139 |
| IMDB-B | Social Networks | 400 | 100 | 100 |
| REDDIT-B | Social Networks | 800 | 200 | 200 |

- **BACE** [11, 33] includes qualitative binding results for a set of inhibitors of human $\beta$-secretase 1, which are potential treatments for Alzheimer's disease. This dataset is used in Alzheimer's disease research and drug development.

- **FreeSolv** [11] includes data on the hydration free energy of small molecules, used for molecular dynamics simulations and solubility studies.

- **Esol** [11] contains data on the aqueous solubility of compounds, used for studying compound solubility and drug design.

- **LIPO** [11] includes data on the lipophilicity of chemical compounds. It is used for studying the partitioning of compounds between water and oil phases, which is important in drug design.

- **MUV** [11, 7] includes data on the activity of compounds from virtual screening, designed for validation of virtual screening techniques.

- **HIV** [11] contains experimentally measured abilities to inhibit HIV replication.

### E.1.2 Protein Datasets

- **PROTEINS** [24] contains data on protein structures and their functionalities. Nodes represent secondary structure elements (SSEs), and edges connect neighboring elements in the amino acid sequence or 3D space. This dataset is used for protein structure prediction and functional analysis.

- **ENZYMES** [24] is a dataset consisting of protein structures classified into enzyme types based on their functionality. It is used for protein function prediction and enzyme classification.

### E.1.3 Social Network Datasets

- **IMDB-BINARY** [24] (abbreviated as IMDB-B) is derived from the collaboration of a movie set. Each graph consists of actors or actresses, with edges representing their cooperation in a movie. The label corresponds to movie's genre. This dataset is used for movie classification and recommendation system studies.

- **IMDB-MULTI** [24] (abbreviated as IMDB-M) consists of graphs derived from movie collaborations which is similar to IMDB-BINARY, but with multi-class labels. It is utilized in multi-class classification tasks in social network analysis.

We also conduct experiments on anomaly detection (AD) settings, where 10 datasets from TU-Dataset [24] are used for evaluation. Following the setting in GlocalKD [21], the samples in the minority class or real anomalous class are viewed as anomalies, while the rest are viewed as normal data. Similar to [21, 51], only normal data are used for model training. Detailed statistics of anomaly detection datasets are compiled in Table 7.

### E.2 Baseline Details

We evaluate the performance of RedOUT by comparing it against 18 state-of-the-art baseline methods. A detailed discussion of these methods is provided below.

- **Non-deep Two-step Methods.** These methods first extract representations using hand-crafted graph kernels and then apply classical OOD detectors. We adopt the Weisfeiler-Lehman (WL) kernel [32] and the propagation kernel (PK) [25] to obtain graph-level representations. On top of these, we apply local outlier factor (LOF) [4], one-class SVM (OCSVM) [22], and isolation forest (iF) [18] for OOD detection.

- **Deep Two-step Methods.** These approaches employ self-supervised graph learning techniques to generate expressive embeddings, followed by a separate OOD detector. We use two representative graph contrastive learning (GCL) methods, InfoGraph [34] and GraphCL [49], to learn representations. For detection, we adopt iF [18] and Mahalanobis distance (MD) [30, 52], both of which have demonstrated effectiveness in prior work.

- **End-to-end Training Methods.** These methods jointly optimize the representation learning and OOD detection objective within a unified framework. We consider GOOD-D [20] as the primary SOTA method in our comparison, which is a GCL-based method that has shown strong performance in unsupervised OOD detection tasks. We also compare our approach with two graph anomaly detection methods that are trained in an end-to-end manner. OCGIN [51], which uses a GIN encoder trained with a support vector data description (SVDD) loss, and GLocalKD [21], which identifies graph anomalies using local-global knowledge distillation.

- **Test-time and Data-centric Methods.** A typical test-time training method is GTrans [12], which adapts representations via test-time contrastive alignment. Since GTrans is not explicitly designed for graph OOD detection, its loss value is utilized as the OOD score. We conduct comparisons based on the experimental results from GOODAT [39]. Data-centric methods leverage well-trained GNN models to fine-tune OOD detectors for identifying OOD samples. These methods mainly include the following approaches: AAGOD [9] employs a graph adaptive amplifier module, which is integrated into a well-trained GNN to facilitate graph OOD detection. Specifically, AAGOD exists in two versions: AAGOD-GIN$_S$+ and AAGOD-GIN$_L$+, corresponding to distinct OOD evaluation methods. GOODAT [39] is the first to directly partition test data into two subgraphs using data-centric techniques in a test-time setting, training a plug-and-play graph masker.

### E.3   Configurations

We conduct the experiments with:

- Operating System: Ubuntu 20.04 LTS.

- CPU: Intel(R) Xeon(R) Gold 6240 CPU @ 2.60GHz, 256GB RAM.

- GPU: Tesla V100 PCIe 32GB GPU.

- Software: Python 3.7, Pytorch 1.8, CUDA 11.0, and Pytorch-Geometric 2.0.1.

### E.4   Additional Experiments Using Structural Entropy as Distinct Metric

By minimizing structural entropy, the structural uncertainty of the graph is reduced, which aids in capturing essential information and identifying distinct patterns between ID and OOD samples. As shown in the score density plots in Figure 1(c), by minimizing structural entropy, the overlap between the representations of ID and OOD graph samples is significantly reduced.

To clarify, we do not treat structural entropy solely as distinct values, but construct a coding tree that reduces redundancy and retains distinctive parts. To further illustrate the performance of using structural entropy directly as distinct values for the OOD detection task, we conduct additional experiments using the 95% range of structural entropy from ID training graphs for OOD detection on test samples. The AUC results shown in Table 8 reveal that using structural entropy directly as a metric causes a significant performance drop. Thus, we can analyze that structural entropy only measures information but does not capture structural differences, and cannot be directly used as an indicator of substantial information. In contrast, our method instantiates substantial information through the encoding tree with minimized structural entropy, achieving good performance.

Table 8: Additional results on using structural entropy as the distinct metric, compared with RedOUT, for OOD detection in terms of AUC (%, mean $\pm$ std).

| ID data
OOD data | BZR
COX2 | PTC-MR
MUTAG | AIDS
DHFR | ENZYMES
PROTEIN | IMDB-M
IMDB-B | Tox21
SIDER | FreeSolv
ToxCast | BBBP
BACE | ClinTox
LIPO | Esol
MUV |
|---|---|---|---|---|---|---|---|---|---|---|
| SE Metric | 51.71±0.60 | 68.29±1.90 | 50.10±0.68 | 54.83±0.97 | 49.07±1.12 | 54.36±0.58 | 47.97±1.36 | 48.97±0.81 | 45.07±0.66 | 52.39±0.60 |
| SEGO | 96.66±0.91 | 85.02±0.94 | 99.48±0.11 | 64.42±4.95 | 80.27±0.92 | 66.67±0.82 | 90.95±1.93 | 87.55±0.13 | 78.99±2.81 | 94.59±0.94 |
| RedOUT | 95.06±0.54 | 94.45±1.66 | 99.98±0.16 | 66.75±2.02 | 79.54±0.72 | 71.67±0.50 | 92.97±0.84 | 92.60±0.23 | 86.56±0.76 | 95.00±0.54 |

## E.5 Comparison Results of RedOUT with Structural Entropy Guided End-to-end Training

SEGO [10] is a recent approach that leverages structural entropy for unsupervised OOD detection. In this section, we conduct a detailed comparison between our method and SEGO to highlight their methodological differences and performance characteristics.

**Difference in Settings.** Firstly, the key differences between our RedOUT and SEGO lie in their settings: RedOUT is designed to improve OOD detection at test-time without modifying pre-trained models or requiring training data, whereas SEGO is an end-to-end unsupervised method that trains from scratch. It is evident that the test-time setting is more practical for real-world applications.

**Difference in Techniques.** Regarding technical details, although both RedOUT and SEGO leverage structural entropy to construct coding trees, their implementations and focal points differ significantly. Specifically, SEGO merely discovers the phenomenon that minimizing structural entropy is beneficial for the OOD detection task. In contrast, our RedOUT's main contribution is **the first to extend GIB to redundancy-aware disentanglement by proposing ReGIB**, which effectively eliminates redundancy and captures essential information through theoretically grounded upper and lower bound optimization.

**Superior Performance of RedOUT.** We further conduct a comparison experiment between RedOUT and SEGO, as shown in Table 8, where our method outperforms SEGO on 8 out of 10 ID/OOD dataset pairs.

## E.6 Efficiency Analysis and Runtime Comparison

Given a graph $G = (\mathcal{V}, \mathcal{E})$, the time complexity of coding tree construction is $O(h(|\mathcal{E}| \log |\mathcal{V}| + |\mathcal{V}|))$, in which $h$ is the height of coding tree after the first step. In general, the coding tree tends to be balanced in the process of structural entropy minimization, thus, $h$ will be around $\log |\mathcal{V}|$. Furthermore, a graph generally has more edges than nodes, *i.e.*, $|\mathcal{E}| \gg |\mathcal{V}|$, thus the runtime almost scales linearly in the number of edges.

**Discussion on the Runtime Comparison.** We compared RedOUT with GOODAT [39] in terms of inference time on test data and the time consumption for coding tree construction (abbreviated as Tree Constr.). For deep learning methods under other settings, we select the best-performing methods, GOOD-D and GraphCL-MD. Shown in Table 9, the time overhead of RedOUT is comparable with the baseline while achieving enhanced performance. Note that coding tree construction is a one-time preprocessing step, it does not impact the efficiency of test-time inference.

The difference in speed is related to the graph density. Specifically, since GOODAT relies on training a graph masker on test samples to separate subgraphs, the iterative optimization of the masker and representation learning of the two subgraphs become frequent and time-consuming when dealing with a large amount of edges in dense graphs. In contrast, RedOUT does not involve a masking design, and the preprocessing step of coding tree construction is non-trainable. For dense social network graphs such as IMDB-M/B, the higher the graph density, the more pronounced the efficiency advantage of RedOUT compared to GOODAT. Detailed dataset statistics can be found in Table 6.

Table 9: Time consumption on pre-training, coding tree construction and test-time OOD detection.

| ID dataset
OOD dataset | BZR
COX2 | PTC-MR
MUTAG | ENZYMES
PROTEIN | IMDB-M
IMDB-B | FreeSolv
ToxCast |
|---|---|---|---|---|---|
| Pre-training (s) | 53.21 | 40.99 | 30.18 | 8.12 | 70.84 |
| Tree Constr. (s) | 0.14 | 0.04 | 0.15 | 0.09 | 0.38 |
| GraphCL-MD (s) | 27.94 | 21.71 | 14.43 | 10.64 | 40.02 |
| GOOD-D (s) | 323.36 | 320.77 | 137.5 | 33.59 | 356.44 |
| GOODAT (s) | 6.86 | 3.55 | 5.24 | 0.87 | 4.81 |
| RedOUT (s) | 7.51 | 5.65 | 2.88 | 0.28 | 7.51 |

**Discussion on the Efficiency of Trainable Parameters.** We further analyze the number of parameters required during training for both RedOUT and GOODAT on specific datasets. RedOUT only updates the parameters of tree encoder. Assuming the encoder has $l$ layers, with each layer consisting of a two-layer MLP with hidden dimension $hid$, the total number of trainable parameters is approximately $2 \times l \times hid \times hid$. In contrast, GOODAT[5] updates both a feature masker and an edge masker, with a parameter size of approximately $batch\_size \times (n \times d + 2 \times m)$, where n is the number of nodes, m is the number of edges, and d is the feature dimension. The edge count is doubled because reciprocal edges are automatically added for undirected graphs.

As shown in Table 6, dataset pairs such as IMDB-M/IMDB-B and ENZYMES/PROTEINS have higher average node and edge counts. Taking IMDB-M/IMDB-B as an example, whose average number of nodes is about 19, and the average number of edges is about 81. With a batch size of 128, GOODAT requires around $128 \times (19 \times 1 + 2 \times 81) = 23,168$ parameters. In comparison, RedOUT uses up to a 5-layer encoder (note that the analysis in Section 5.5 shows that 5 layers are not strictly necessary for optimal performance) with hidden dimension 32, requiring $2 \times 5 \times 32 \times 32 = 10,240$ trainable parameters. Thus, for dense graphs with more edges on average, RedOUT requires fewer parameters and less runtime. However, for sparser graphs such as PTC_MR/MUTAG, GOODAT's parameter count is $128 \times (16 \times 1 + 2 \times 17) = 6,400$, which is smaller than that of RedOUT. Therefore, GOODAT is more efficient on sparse graphs.

### E.7 Additional Results of Parameter Study on OOD Detection

We illustrate additional results of hyperparameter sensitivity analysis on OOD detection datasets in Figure 14 and Figure 15. We provide additional analysis here.

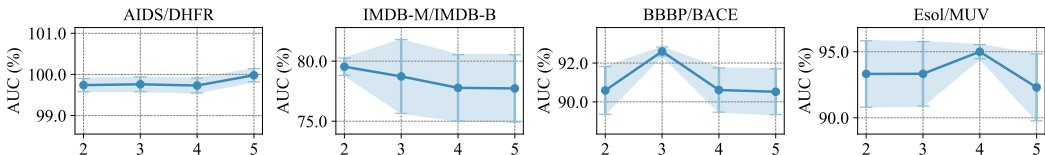

Figure 14: Additional results of the natural hierarchy of graph on OOD detection.

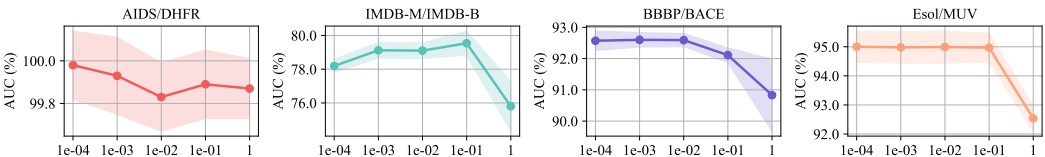

Figure 15: Additional results of the sensitivity of hyperparameter $\lambda$ on OOD detection.

**Height $k$ of Graph's Natural Hierarchy on OOD Detection.** We observe that the impact of coding tree height on OOD detection performance in Figure 14 varies slightly across different datasets. Specifically, height $k$ can be interpreted as the number of hierarchical aggregations of essential information. Corresponding to the number of layers in GNNs, the range of $k$ values we selected is from 2 to 5. The optimal value of $k$ differs due to the varying graph structures of different datasets, yet it fluctuates within a reasonable range. The current coding tree optimization algorithm relies on the fixed height applied to the whole dataset, without considering the diversity among samples. One potential direction for improving our method is to incorporate an adaptive coding tree height to better extract essential structures in graphs, which we leave as our future work.

**Sensitivity Analysis of $\lambda$ on OOD Detection.** To analyze the sensitivity of $\lambda$ for RedOUT, we alter the value from 1e-04 to 1. Results in Figure 15 demonstrate the performance is sensitive to changes in $\lambda$ and contains a reasonable range across different datasets.

---

[5]Source code of GOODAT is available at `https://github.com/ee1s/goodat`.

Table 10: Anomaly detection results in terms of AUC (%, mean ± std). The best and runner-up results are highlighted with **bold** and underline, respectively.

| ID Dataset | PROTEINS-full | ENZYMES | AIDS | DHFR | BZR | COX2 | DD | NCI1 | IMDB-B | REDDIT-B | A.A | A.R. |
|---|---|---|---|---|---|---|---|---|---|---|---|---|
| PK-OCSVM | 50.49±4.92 | 53.67±2.66 | 50.79±4.30 | 47.91±3.76 | 46.85±5.31 | 50.27±7.91 | 48.30±3.98 | 49.90±1.18 | 50.75±3.10 | 45.68±2.24 | 49.46 | 10.6 |
| PK-iF | 60.70±2.55 | 51.30±2.01 | 51.84±2.87 | 52.11±3.96 | 55.32±6.18 | 50.05±2.06 | 71.32±2.41 | 50.58±1.38 | 50.80±3.17 | 46.72±3.42 | 54.07 | 9.1 |
| WL-OCSVM | 51.35±4.35 | 55.24±2.66 | 50.12±3.43 | 50.24±3.13 | 50.56±5.87 | 49.86±7.43 | 47.99±4.09 | 50.63±1.22 | 54.08±5.19 | 49.31±2.33 | 50.94 | 9.7 |
| WL-iF | 61.36±2.54 | 51.60±3.81 | 61.13±0.71 | 50.29±2.77 | 52.46±3.30 | 50.27±0.34 | 70.31±1.09 | 50.74±1.70 | 50.20±0.40 | 48.26±0.32 | 54.66 | 8.9 |
| GraphCL-iF | 60.18±2.53 | 53.60±4.88 | 79.72±3.98 | 51.10±2.35 | 60.24±5.37 | 52.01±3.17 | 59.32±3.92 | 49.88±0.53 | 56.50±4.90 | 71.80±4.38 | 59.43 | 8.0 |
| OCGIN | 70.89±2.44 | 58.75±5.98 | 78.16±3.05 | 49.23±3.05 | 65.91±1.47 | 53.58±5.05 | 72.27±1.83 | **71.98±1.21** | 60.19±8.90 | 75.93±8.65 | 65.69 | 5.9 |
| GLocalKD | 77.30±5.15 | 61.39±8.81 | 93.27±4.19 | 56.71±3.57 | 69.42±7.78 | 59.37±12.67 | **80.12±5.24** | 68.48±2.39 | 52.09±3.41 | 77.85±2.62 | 69.60 | 4.2 |
| GOOD-D$_{simp}$ | 74.74±2.28 | 61.23±4.58 | 94.09±1.75 | 62.71±3.38 | 74.48±4.91 | 60.46±12.34 | 72.24±1.82 | 59.56±1.62 | 65.49±1.06 | 87.87±1.38 | 71.29 | 3.8 |
| GOOD-D | 71.97±3.86 | 63.90±3.69 | **97.28±0.69** | 62.67±3.11 | 75.16±5.15 | **62.65±8.14** | 73.25±3.19 | 61.12±2.21 | 65.88±0.75 | 88.67±1.24 | 72.26 | 2.7 |
| GTrans | 60.16±5.06 | 38.02±6.24 | 84.57±1.91 | 61.15±2.87 | 51.97±8.15 | 53.56±3.47 | 76.73±2.83 | 41.42±2.16 | 45.34±3.75 | 69.71±2.21 | 58.26 | 8.5 |
| GOODAT | **77.92±2.37** | 52.33±4.74 | 95.50±0.99 | 61.52±2.86 | 64.77±3.87 | 59.99±9.76 | 77.62±2.88 | 45.96±2.42 | 65.46±4.34 | 80.31±0.85 | 68.14 | 4.8 |
| RedOUT | 76.08±3.93 | **77.64±5.64** | 96.53±0.61 | **65.51±4.95** | **89.62±4.72** | 61.49±5.14 | 74.61±3.45 | 71.34±1.22 | **67.48±0.59** | **89.81±2.54** | **77.01** | **1.8** |

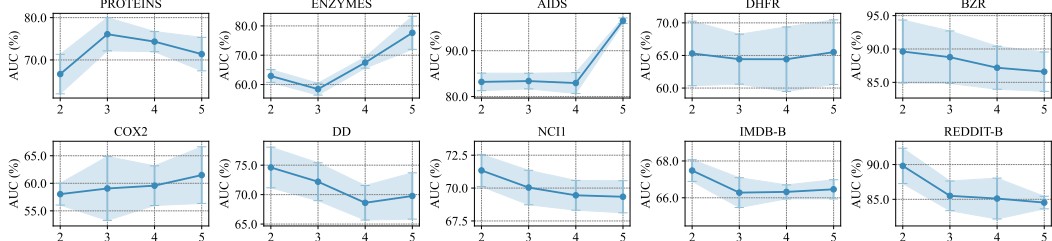

Figure 16: The natural hierarchy of graph on AD datasets.

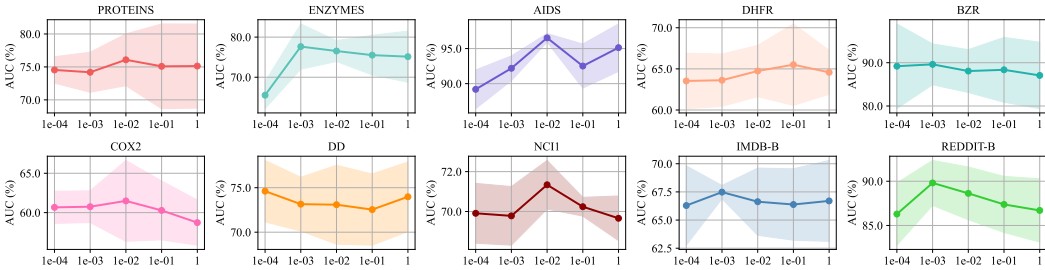

Figure 17: The sensitivity of hyperparameter $\lambda$ on AD datasets.

## E.8 Additional Results of Anomaly Detection Performance

To investigate if RedOUT can generalize to the anomaly detection setting [51, 21], we conduct experiments on 10 datasets following the benchmark in GLocalKD and GOOD-D [21, 20], where only normal data are used for model training. The results are shown in Table 10. From the results, we find that our method achieves the best performance on 5 out of 10 datasets and runner-up performance on 3 datasets, with an average improvement of 4.75% over the SOTA. This demonstrates that RedOUT indeed has a strong capability to transfer to anomaly detection tasks. Thus, we can conclude that capturing common patterns in the anomaly detection setting is crucial, which is directly reflected in the performance.

## E.9 Additional Results of Parameter Study on Anomaly Detection

We also illustrate additional results of hyperparameter sensitivity analysis on anomaly detection datasets in Figure 16 and Figure 17. We provide additional analysis here.

**Height $k$ of Graph's Natural Hierarchy on Anomaly Detection.** We also conducted experiments on the impact of the coding tree height $k$ on 10 anomaly detection datasets, as shown in Figure 16. We observe that the impact of coding tree height on anomaly detection performance varies slightly across different datasets.

**Sensitivity Analysis of $\lambda$ on Anomaly Detection.** To analyze the sensitivity of $\lambda$ for RedOUT, we alter the value from 1e-04 to 1. The AUC w.r.t different selections of $\lambda$ is plotted in Figure 17.

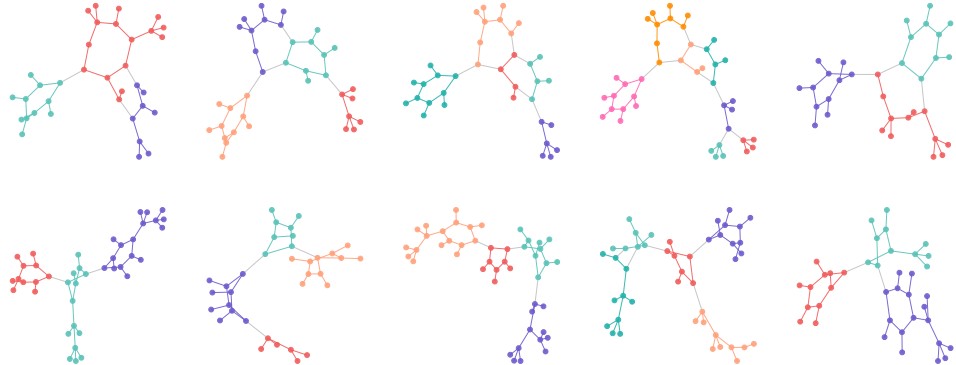

Figure 18: Visualizing of the essential structure based on the coding tree preserved by RedOUT on BZR/COX2, where the first row is BZR and the second row is COX2.

Results demonstrate the performance is sensitive to changes in $\lambda$ and contains a reasonable range across different datasets.

### E.10 Additional Case Study on ID/OOD Dataset Pairs

In this additional case study, we further demonstrate the effectiveness of our method in extracting the intrinsic structural features of graphs. We visualize the graph structures extracted by the encoding tree based on minimizing structural entropy on 10 pairs of ID/OOD datasets, as shown from Figure 18 to Figure 27.

In these visualizations, the colored nodes represent different communities based on the subtree structures of the coding tree. The edges marked with different colors are those involved in the merging process during the construction of the coding tree, with each color indicating a different subtree partition. We provide additional analysis as follows. RedOUT effectively extracts the distinct intrinsic structures of ID and OOD graph data in datasets related to molecules, proteins, and social networks, which is crucial for OOD detection. For graphs in social networks, which typically exhibit high connectivity and edge density, incorporating node and edge attributes implies huge potential to enhance the model's representational capacity, beyond the essential structural information we currently extract. However, the current IMDB-M/IMDB-B datasets do not incorporate node or edge attributes. This is a promising direction that inspires us to take a deep exploration in our future work. As illustrated in Figure 22, although the similarity in social network structures leads to suboptimal performance on the IMDB-M/IMDB-B datasets, it still captures the differences in intrinsic structures. Moreover, our method successfully extracts distinctive structures in the other datasets.

## F  Broader Impacts

Graph OOD detection contributes to improving the robustness of GNNs in safety-critical applications and scientific discovery tasks such as drug design and protein interaction analysis. It is important to ensure that such techniques are used to enhance reliability rather than to analyze user behavior without proper authorization.

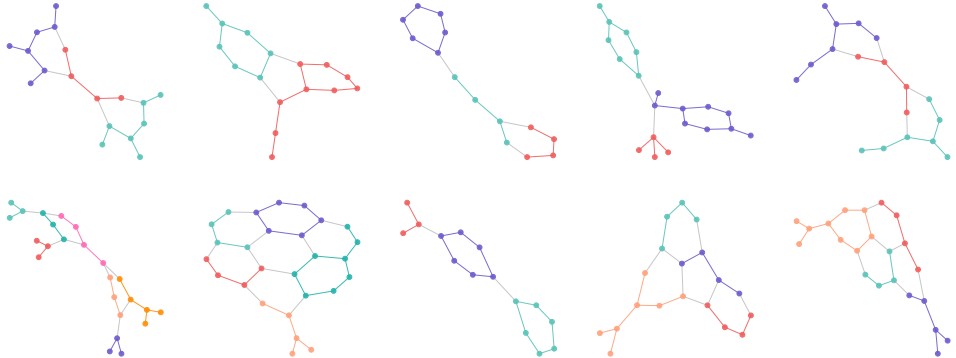

Figure 19: Visualizing of the essential structure based on the coding tree preserved by RedOUT on PTC-MR/MUTAG, where the first row is PTC-MR and the second row is MUTAG.

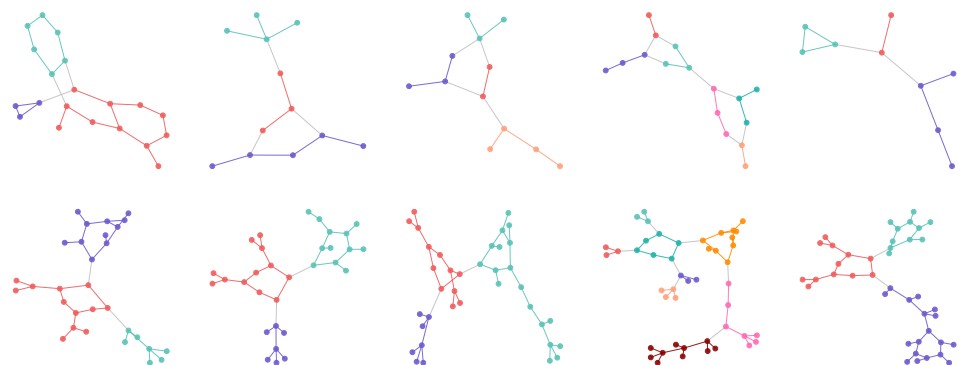

Figure 20: Visualizing of the essential structure based on the coding tree preserved by RedOUT on AIDS/DHFR, where the first row is AIDS and the second row is DHFR.

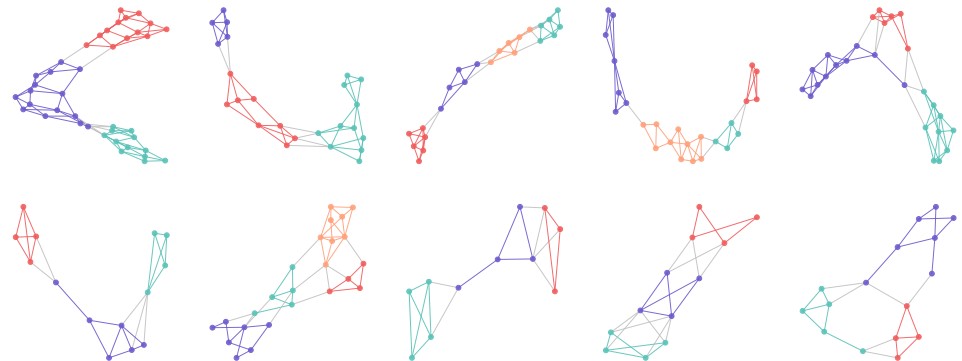

Figure 21: Visualizing of the essential structure based on the coding tree preserved by RedOUT on ENZYMES/PROTEIN, where the first row is ENZYMES and the second row is PROTEIN.

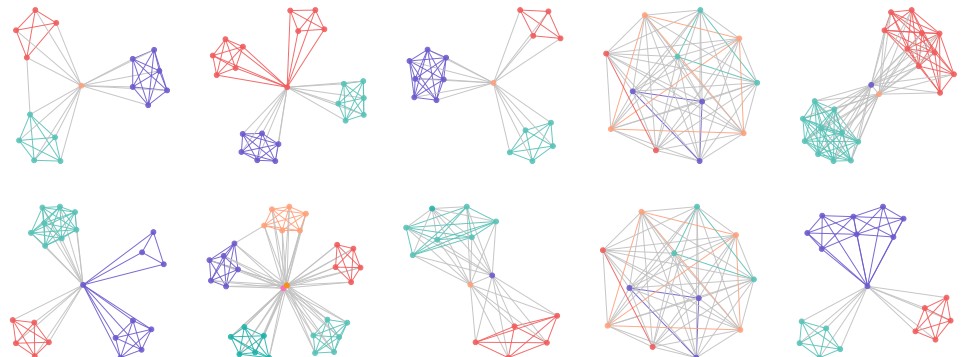

Figure 22: Visualizing of the essential structure based on the coding tree preserved by RedOUT on IMDB-M/IMDB-B, where the first row is IMDB-M and the second row is IMDB-B.

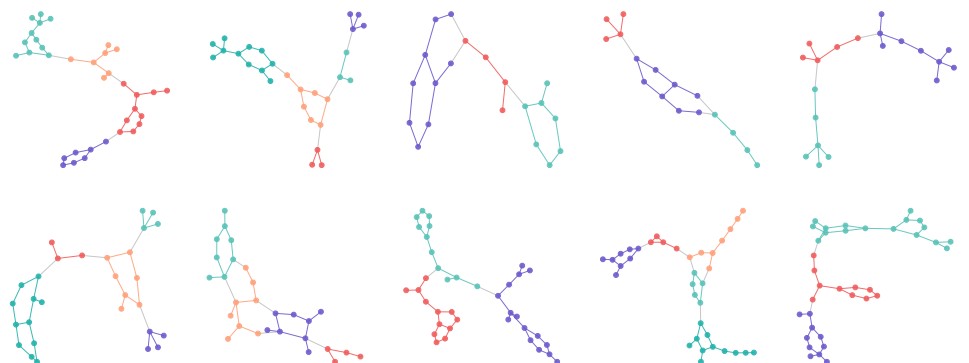

Figure 23: Visualizing of the essential structure based on the coding tree preserved by RedOUT on Tox21/SIDER, where the first row is Tox21 and the second row is SIDER.

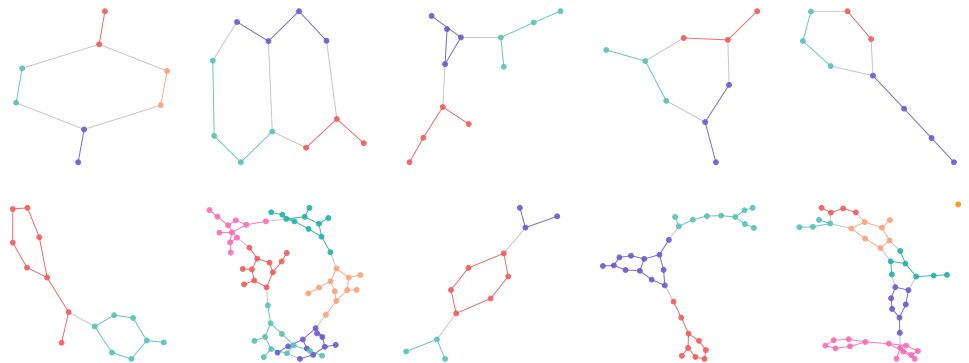

Figure 24: Visualizing of the essential structure based on the coding tree preserved by RedOUT on FreeSolv/Toxcast, where the first row is FreeSolv and the second row is Toxcast.

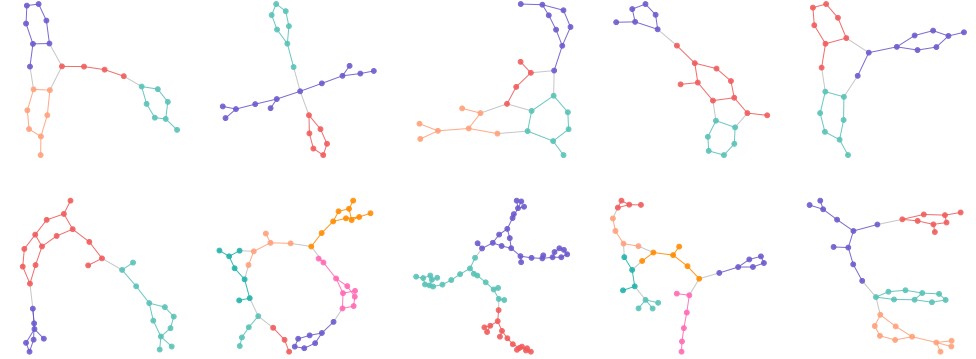

Figure 25: Visualizing of the essential structure based on the coding tree preserved by RedOUT on BBBP/BACE, where the first row is BBBP and the second row is BACE.

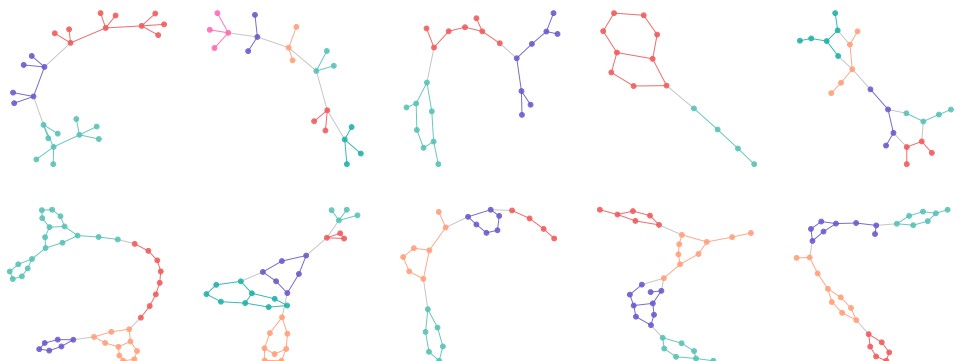

Figure 26: Visualizing of the essential structure based on the coding tree preserved by RedOUT on ClinTox/LIPO, where the first row is ClinTox and the second row is LIPO.

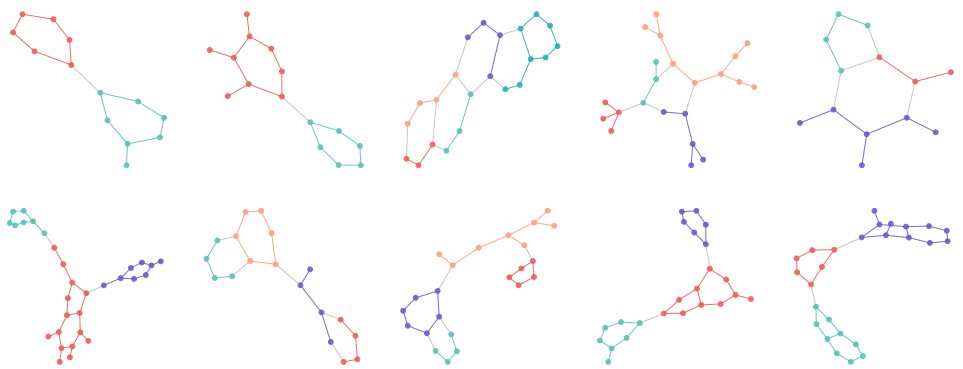

Figure 27: Visualizing of the essential structure based on the coding tree preserved by RedOUT on Esol/MUV, where the first row is Esol and the second row is MUV.

