# OpenReview forum: "Redundancy-Aware Test-Time Graph Out-of-Distribution Detection"
_NeurIPS.cc/2025/Conference — NeurIPS 2025 poster_

### Official Review · Reviewer_tiWp · 2025-06-30

**Clarity:** 1
**Significance:** 2
**Originality:** 3
**Rating:** 4
**Confidence:** 3

**Summary:**

This paper proposes to construct a coding tree based on a well-trained GNN to reduce the redundancy in the graph structure and use the construction loss as a criterion for OOD detection.

**Questions:**

* Since the proposed method uses the optimized loss as an OOD detection score, how does the method identify a sample as an OOD sample? Is there a manually set threshold that separates IID and OOD samples?

* While the paper provides time complexity analysis, could the comparison between the proposed method and other baseline methods be provided? Constructing and optimizing a coding tree can be a time-consuming process. It would be beneficial to have more details about optimizing the coding tree on a sample provided.

I look forward to the authors' response and will adjust my rating based on the rebuttal.

**Ethical Concerns:**

["NO or VERY MINOR ethics concerns only"]

**Final Justification:**

This paper proposes a new OOD detection method using a coding tree. The proposed method seems novel. My initial concerns has been addressed in the rebuttal, therefore I am increasing my rating.

**Limitations:**

The authors have discussed the limitation that the proposed method mainly focuses on graph-level detection. However, I encourage the authors to further discuss the limitations of the proposed method.

**Paper Formatting Concerns:**

I do not notice any major formatting issues in this paper.

**Quality:**

2

**Strengths And Weaknesses:**

**Strength**:
* The proposed method seems novel. This paper proposes to detect OOD samples by constructing a coding tree that minimizes the structural entropy.

* The paper provides sufficient experimental results to verify the effectiveness and sensitivity to hyperparameter settings of the proposed method.

**Weakness**:
* Some parts of the paper are hard to follow. For example, Figure 2 contains "?", which makes it difficult to understand. Sec. 3 introduces the ReGIB and the objective for coding tree construction, but lacks a straightforward explanation on the choice of using the construction loss as the OOD detection score.

* The details of how the OOD samples are detected are vague. While constructing the coding tree provides a loss that is used as an OOD detection score, the method for separating OOD samples from IID samples seems unclear. If the OOD samples are detected when the loss exceeds a certain threshold, the good performance of the proposed method may stem from a well-tuned threshold on the benchmark datasets.

* The paper could benefit from more empirical details on the coding tree construction and a time consumption comparison between the proposed method and other baseline methods.

---

> ### Author Rebuttal · Authors · 2025-07-30
>
> We sincerely appreciate your valuable comments and will reply to all these concerns individually.
>
> **W1.1:** Figure 2 contains "?", which makes it difficult to understand
>
> We would like to clarify that Figure 2 in our paper does not contain "?" symbols. The issue you mentioned may result from incomplete image rendering in the browser. We recommend downloading the paper and viewing it locally using Adobe Reader to ensure correct visualization.
>
> ---
> **W1.2 & W2.1:** Section 3 introduces ReGIB and the objective for coding tree construction, but lacks a straightforward explanation on the choice of using the construction loss as the OOD detection score
>
> Thank you for the question. The construction of coding tree does not introduce any new loss function. Instead, we use structural entropy as the optimization metric for coding tree construction (see Eq. (12) and (14)).
> Moreover, we would like to clarify that we do not use structural entropy as the OOD score, and the OOD score is actually computed based on Eq. (18).
>
> ---
> **W2.2:** If the OOD samples are detected when the loss exceeds a certain threshold, the good performance of the proposed method may stem from a well-tuned threshold on the benchmark datasets
>
> Thank you for the comment. The strong performance of our method does not stem from a well-tuned threshold.
> We follow the same evaluation protocol as GOOD-D and GOODAT, where the AUC metric is computed using `auc = sklearn.metrics.roc_auc_score(y, ood_score)`.
> Specifically, `y` denotes the binary ground-truth labels, and `ood_score` represents the confidence score for predicting a sample as OOD. This score is a continuous value that can be ranked across samples. AUC is computed by ranking all samples according to their `ood_score`, and then calculating the corresponding true positive rate (TPR) and false positive rate (FPR) over the full range of score distributions to construct the ROC curve (with FPR on the x-axis and TPR on the y-axis). The AUC is the area under this ROC curve, with a larger value indicating stronger discrimination ability. The effectiveness of these methods, including ours, is thus evaluated independently of any manually selected threshold.
>
> ---
> **W3 & Q2.2:** The paper could benefit from more empirical details on the coding tree construction and a time consumption comparison between the proposed method and other baseline methods
>
> Thank you for pointing this out. We provide detailed illustrations and explanations of the coding tree construction process in Appendix D and Figures 10 to 12 of the paper.
> Moreover, coding tree construction, as well as the proposed RedOUT method, is not time-consuming. In Appendix E7, Table 10 compares the time consumption of pretraining, coding tree construction, and test-time OOD detection.
>
> We additionally provide a comparison of time consumption with other baselines in the table below. For deep learning methods under other settings, we select the best-performing methods, GOODD and GraphCL-MD, for reference.
>
> |ID|BZR| PTC-MR|ENZYMES| IMDB-M|FreeSolv|
> |-|-|-|-|-|-|
> |OOD| COX2   | MUTAG  | PROTEIN | IMDB-B | ToxCast  |
> |Coding Tree Construction (s) | 0.14   | 0.04   | 0.15    | 0.09   | 0.38     |
> |GraphCL-MD (s)| 27.94  | 21.71  | 14.43   | 10.64  | 40.02|
> |GOODD (s)| 323.36 | 320.77 | 137.5   | 33.59  | 356.44|
> |GOODAT (s)| 6.86   | 3.55   | 5.24    | 0.87   | 4.81|
> |RedOUT (s)| 7.51   | 5.65   | 2.88    | 0.28   | 7.51|
>
> Note that the official implementation of AAGOD could not run successfully in our experimental environment. Several issues have also been reported by other users on its GitHub repository and remains unresolved.
>
> ---
> **Q1:** Since the proposed method uses the optimized loss as an OOD detection score, how does the method identify a sample as an OOD sample? Is there a manually set threshold that separates IID and OOD samples?
>
> Thank you for the insightful question. A global evaluation metric AUC is used to assess a model’s ability to distinguish between ID and OOD samples overall. In practical applications, we can estimate a decision threshold based on the ID distribution observed in the pretraining dataset. For example, a score threshold that corresponds to a 5% false positive rate (FPR) on ID samples can be chosen. This threshold can then be used at test time to decide whether a new input is OOD. Notably, the selection of threshold does not affect the evaluation of the model’s OOD detection performance, that is, the threshold selection does not affect the computation of AUC metric.
>
> ---
> **Q2.1:** While the paper provides time complexity analysis, could the comparison between the proposed method and other baseline methods be provided?
>
> Thank you for the valuable question. A detailed analysis and comparison of computational complexity between our proposed RedOUT and GOODAT is provided on page 31 in Appendix E7.
> In addition, we provide a comparative analysis of time complexity for our proposed method and several other representative baselines as follows:
>
> - **RedOUT** only optimizes a tree encoder with $L$ layers, each being a two-layer MLP with hidden dimension $h$. The total time complexity is $O(L(mh + nh^2))$ per graph.
> - **GOOD-D** updates two GIN encoders with $L$ layers and hidden size $h$. The forward pass complexity is $O(L(mh + nh^2))$ per graph. An additional $O(nh)$ cost arises from the group-space clustering, resulting in $O(L(mh + nh^2)+nh)$ overall.
> - **GTrans** performs optimization over node features and edge structure based on a fixed backbone. The per-graph time complexity is $O(L(mh + nh^2)+md + nd^2 + B'\log B')$, where $B'$ is the number of candidate edge flips.
> - **AAGOD** introduces an auxiliary augmentation scorer alongside the main GNN model with $L$ layers and hidden dimension $h$ that scoring $K$ augmentations per graph. The total time complexity is $O(L(mh + nh^2)+K(nd+m))$.
> - **GOODAT** introduces two masking modules (feature and edge maskers). The time complexity per graph is $O(L(mh + nh^2)+nd + m)$, as it applies learnable masks over node features and edge connections.
>
> Overall, the time complexity of our proposed RedOUT is comparable to that of other baselines.
>
> ---
> **Limitation:** The authors have discussed the limitation that the proposed method mainly focuses on graph-level detection. However, I encourage the authors to further discuss the limitations of the proposed method.
>
> Thank you for the suggestion. The current coding tree optimization algorithm relies on the fixed height applied to the whole dataset, without considering the diversity among samples. One potential direction for improving our method is to incorporate an adaptive coding tree height to better extract essential structures in graphs, which we leave as our future work. We will expand the discussion of limitations in the revised version.
>
> We once again thank you for your constructive feedback and valuable suggestions. We will carefully address all the raised points in the revised version.

---

> > ### Comment · Reviewer_tiWp · 2025-08-04
> >
> > I appreciate the rebuttal provided by the authors, which has addressed most of my concerns. I will increase my rating.

---

> > > ### Author Response · Authors · 2025-08-04
> > >
> > > Dear Reviewer  tiWp,
> > >
> > > We would like to express our sincere gratitude to you for endorsing our work and providing constructive suggestions. Thanks again for the time and effort in reviewing our work!

---

### Official Review · Reviewer_HHWR · 2025-07-02

**Clarity:** 3
**Significance:** 3
**Originality:** 2
**Rating:** 4
**Confidence:** 2

**Summary:**

The paper proposes RedOUT, a framework for test-time graph out-of-distribution (OOD) detection that addresses the challenge of structural redundancy in graph data. It introduces the Redundancy-aware Graph Information Bottleneck (ReGIB) to separate essential information from irrelevant redundancy by minimizing structural entropy, which helps in effectively distinguishing between in-distribution and out-of-distribution graph samples. The framework demonstrates superior performance over state-of-the-art baselines through extensive experiments on real-world datasets, highlighting its capability to capture essential structural information in graphs.

**Questions:**

1. It lacks clear introduction of the baseline to analyze the contribution of this work.
2. The performance of RedOUT is sensitive to hyperparameters such as the coding tree height k and the trade-off parameter λ. While the paper provides a reasonable range for these parameters, it lacks detailed analysis on how to select optimal values for different datasets, which may affect the method's practical usability.
3. The coding tree construction process involves structural entropy minimization, which may incur higher computational costs for large-scale graphs.

**Ethical Concerns:**

["NO or VERY MINOR ethics concerns only"]

**Final Justification:**

The paper proposes RedOUT, a framework for test-time graph out-of-distribution (OOD) detection that addresses the challenge of structural redundancy in graph data. It introduces the Redundancy-aware Graph Information Bottleneck (ReGIB) to separate essential information from irrelevant redundancy by minimizing structural entropy, which helps in effectively distinguishing between in-distribution and out-of-distribution graph samples. The framework demonstrates superior performance over state-of-the-art baselines through extensive experiments on real-world datasets, highlighting its capability to capture essential structural information in graphs.
By minimizing structural entropy, it effectively extracts essential structural information from graphs while eliminating redundancy, offering a new perspective for test-time graph OOD detection.
Extensive experiments on multiple real-world graph datasets demonstrate that RedOUT outperforms state-of-the-art baselines, achieving an average improvement of 6.7% in unsupervised OOD detection tasks.
The paper provides theoretical analysis and proofs for ReGIB, such as deriving upper and lower bounds for essential and redundant information optimization.
The Paper is well written and easy to follow.

**Limitations:**

1. The coding tree construction process involves structural entropy minimization, which may incur higher computational costs for large-scale graphs.

**Quality:**

3

**Strengths And Weaknesses:**

Strength:
1. By minimizing structural entropy, it effectively extracts essential structural information from graphs while eliminating redundancy, offering a new perspective for test-time graph OOD detection.
2. Extensive experiments on multiple real-world graph datasets demonstrate that RedOUT outperforms state-of-the-art baselines, achieving an average improvement of 6.7% in unsupervised OOD detection tasks.
3. The paper provides theoretical analysis and proofs for ReGIB, such as deriving upper and lower bounds for essential and redundant information optimization.
4. The Paper is well written and easy to follow.

Weakness:
1. It lacks clear introduction of the baseline to analyze the contribution of this work.
2. The performance of RedOUT is sensitive to hyperparameters such as the coding tree height k and the trade-off parameter λ. While the paper provides a reasonable range for these parameters, it lacks detailed analysis on how to select optimal values for different datasets, which may affect the method's practical usability.
3. The coding tree construction process involves structural entropy minimization, which may incur higher computational costs for large-scale graphs.

---

> ### Author Rebuttal · Authors · 2025-07-30
>
> We sincerely appreciate your thoughtful comments and address your comments and suggestions point by point below.
>
> **Q1:** Further analysis of baselines and the contribution of this work
>
> Thank you for the suggestion.
> In Appendix E2, we provide a detailed categorization and discussion of the 18 baseline methods considered in our comparison. These baselines are grouped into 4 categories: Non-deep Two-step Methods, Deep Two-step Methods, End-to-end Training Methods, and Test-time and Data-centric Methods. Additional discussion and comparison with related works are provided in Appendix B. We summarize the key differences and contributions below.
>
> Non-deep two-step methods, deep two-step methods, and end-to-end training methods all rely on training data to modify the parameters of GNN models. Data-centric methods such as AAGOD do not alter pre-trained GNN parameters but still require access to ID training data for post-hoc data augmentation.
> In contrast, our method is designed for the test-time setting, aiming to improve OOD detection performance without modifying pre-trained models or requiring any access to training data. Although Gtrans and GOODAT are also test-time methods, they requires additional training of graph transformation functions or graph maskers. In our case, the extraction of essential structure is achieved via structure entropy minimization, which is a training-free optimization procedure.
> We would like to further emphasize that the core contribution of this work lies in the first to extend GIB framework to redundancy-aware disentanglement. Specifically, we propose ReGIB, which effectively removes redundant information and extracts essential structure by jointly optimizing theoretical upper and lower bounds.
>
> Thanks again and we will include further analysis in the revised version.
>
> ---
> **Q2:** How to select optimal values for different datasets
>
> Thank you for the insightful question.
> For the hyperparameter $k$, while the performance may exhibit slight variations, we observe that RedOUT consistently achieves strong results, as illustrated in Figure 4. Specifically, on 5 out of 6 datasets, RedOUT outperforms all baselines regardless of the choice of $k$. Furthermore, compared with GOODAT (which also operates in test-time setting), our method achieves superior performance on all 6 datasets. These results demonstrate that ReGIB can effectively extract essential structural information and eliminate irrelevant redundancy, regardless of the depth of  coding tree.
> For the hyperparameter $\lambda$, we empirically find that values in the range from 1e-4 to 1e-2 lead to consistently effective performance.
> This robustness reduces the need for extensive hyperparameter tuning, making our method well-suited for practical deployment in real-world scenarios.
>
> ---
> **Q3:** Analysis of the computational overhead of coding tree construction on large-scale graphs
>
> Thank you for the suggestion. We provide a detailed analysis of the scaling efficiency of our method in Appendix E7. As shown in Figure 13, the results indicate that from small-scale graphs to larger ones such as the OGB datasets (e.g., ogbg-code2), the runtime and memory consumption of coding tree construction increase nearly linearly with the number of nodes.
>
> We once again thank you for your constructive feedback and valuable suggestions. We will carefully address all the raised points in the revised version.

---

> > ### Comment · Reviewer_HHWR · 2025-08-08
> >
> > I appreciate the rebuttal provided by the authors, which has addressed some of my concerns. I believe a Borderline Accept score remains fair for this paper. I will therefore maintain my original score.

---

### Official Review · Reviewer_mdJJ · 2025-07-03

**Clarity:** 3
**Significance:** 3
**Originality:** 3
**Rating:** 5
**Confidence:** 3

**Summary:**

This paper proposes RedOUT, a redundancy-aware framework for test-time graph OOD detection. The method introduces a Redundancy-aware Graph Information Bottleneck , which decomposes graph information into essential components and redundant noise. By minimizing structural entropy on a coding tree constructed from test graphs, RedOUT extracts distinctive substructures critical for OOD detection while preserving semantic consistency. The framework operates as a post-hoc, unsupervised test-time algorithm without retraining, achieving substantial improvements on several graph benchmarks. Experimental results demonstrate sota performance  on both OOD detection and anomaly detection tasks.

**Questions:**

1. How robust is RedOUT to incorrect pseudo-labels at test time, especially on highly uncertain OOD samples? Could this reliance on pseudo-labels amplify noise or bias in the essential view extraction?
2. Since RedOUT mainly improves on molecular graph datasets, what specific characteristics of social network graphs limit its performance? Could incorporating node/edge attributes (besides structure) help mitigate this?
3. Given the strong reliance on structural entropy minimization, how sensitive is the framework to the choice of coding tree height k in unseen datasets or new application scenarios? Could an adaptive or data-driven selection of k improve stability?

**Ethical Concerns:**

["NO or VERY MINOR ethics concerns only"]

**Final Justification:**

The authors have addressed most of my concerns, after reviewing the other reviewers' comments, I have decided to maintain my original score.

**Quality:**

4

**Strengths And Weaknesses:**

Strengths:
1. Novel formulation of redundancy-aware test-time OOD detection for graphs, filling a gap in current data-centric, post-hoc graph OOD detection literature.
2. The introduction of ReGIB with theoretical bounds and entropy minimization provides a well-motivated, principled approach for isolating essential information.
3. The ablation studies and parameter sensitivity analyses are thorough and help clarify the contribution of each component.
4. The framework does not require model retraining, which makes it practical for real-world applications.

Weaknesses:
1. The performance improvement of RedOUT on the IMDB datasets appears to be relatively limited. Could the authors provide an explanation for this phenomenon?
2. The reliance on pseudo-labels during test-time introduces potential error accumulation that is not deeply analyzed
3. The time complexity of the coding tree construction is relatively high (O(h|E|log|V|)), which may affect scalability on large graphs.

---

> ### Author Rebuttal · Authors · 2025-07-30
>
> We sincerely appreciate your valuable comments and are happy to address your concerns as follows:
>
> **W1 & Q2:** An explanation for the relatively limited performance improvement on the IMDB dataset. Could incorporating node/edge attributes (besides structure) help?
>
> Thank you for the insightful suggestion. In Section 4.5, through a case study, we investigate that the suboptimal performance mainly stems from the highly similar structural semantic information and the shared data source, which makes it challenging even for existing methods to effectively distinguish OOD samples.
>
> For graphs in social networks, which typically exhibit high connectivity and edge density, incorporating node and edge attributes implies huge potential to enhance the model’s representational capacity, beyond the essential structural information we currently extract.
> However, the current IMDB-B/M datasets do not incorporate node or edge attributes. This is a promising direction that inspires us to take a deep exploration in our future work. We will include further analysis in the revision.
>
>
> ---
> **W2 & Q1.1:** The impact of incorrect pseudo-labels at test time on RedOUT, especially for highly uncertain OOD samples
>
> Thank you for the valuable question.
> While the pseudo-labels predicted by pre-trained model may contain errors when encountering highly uncertain OOD samples, such errors do not compromise the effectiveness of our redundancy removal mechanism. Specifically, we leverage ReGIB to disentangle the conditional redundant information associated with pseudo-labels in Eq. (6), and optimize the upper bound in Eq. (8) to minimize $I(f(G ^ *); f(G) \mid \tilde{Y})$.
>
> In the general case, we decompose the learned representation as $f(G) \rightarrow{R_1 + Y_G}$ and $f(G ^ *) \rightarrow{R_2 + Y_G}$, where $R_{1/2}$ denotes label-irrelevant redundant information, and $Y_G$ denotes label-related information.
> Minimizing $I(f(G ^ *); f(G) \mid \tilde{Y})$ is equivalent to minimizing $I((R_1+Y_G);(R_2+Y_G) \mid \tilde{Y}) \rightarrow \min I(R_1; R_2)$.
>
> If the pseudo-label $\tilde{Y}$ predicted by the pre-trained model is incorrect, then $f(G) \mid \tilde{Y} \rightarrow \bar{R}_1 + \bar{Y}_G$ and $f(G ^ *) \mid \tilde{Y} \rightarrow \bar{R}\_2 + \bar{Y}\_G$,
> where $ \bar{R}\_{1/2}$ denotes the residual irrelevant information after removing the wrongly associated label part, and $\bar{Y}\_G$ similarly denotes the remaining label-related information.
> In this case, minimizing $I(f(G ^ *); f(G) \mid \tilde{Y})$ becomes $\min I(\bar{R}_1 + \bar{Y}_G ; \bar{R}_2 + \bar{Y}_G) \rightarrow \min I(\bar{R}_1 ; \bar{R}_2)$.
>
> Hence, the optimization still targets the irrelevant redundancy and the direction of our objective remains unchanged. Therefore, the effectiveness of our method in removing irrelevant redundancy is not compromised.
>
> ---
> **W3:** Scalability of coding tree construction on large graphs
>
> Thanks for your suggestions. We provide a scalability efficiency analysis of our method in Appendix E7. As shown in Figure 13, the results illustrate that from smaller scales to the OGB datasets (e.g., ogbg-code2), the runtime and memory usage scale up nearly linearly with the number of nodes. Furthermore, as shown in Table 10, the time overhead introduced by coding tree construction is negligible compared to the cost of model pretraining and test-time detection.
>
> ---
> **Q1.2:** Could the reliance on pseudo-labels amplify noise or bias in the essential view extraction?
>
>  Thanks for the question. The extraction of essential information does not depend on pseudo-labels and thus is not subject to such bias. It is derived independently by minimizing the structure entropy in Eq. (12). We will include the additional explanation in the revision.
>
> ---
> **Q3:** How sensitive is the framework to the choice of coding tree height $k$ in unseen datasets or new application scenarios? Could an adaptive or data-driven selection of k improve stability?
>
> Thanks for the question. The impact of the coding tree height $k$ on performance is analyzed in Figure 4. We observe that across 5 out of 6 datasets, our method outperforms the baselines regardless of the specific choice of $k$. Notably, compared with GOODAT, which also operates in a test-time setting, our method achieves superior performance across all 6 datasets. This further confirms the effectiveness of our proposed ReGIB in extracting essential structures and removing redundancy, irrespective of the coding tree depth.
>
> Additionally, in light of the variability of test-time samples, an adaptive or data-driven selection of $k$ shows great potential for improving stability. We leave this exploration for future work.
>
>
> We once again thank you for your constructive feedback and valuable suggestions. We will carefully address all the raised points in the revised version.

---

### Note · Authors · 2025-08-13

Dear Reviewers and ACs,

We sincerely thank all reviewers for their detailed and constructive feedback on our manuscript. We are delighted that they acknowledge our contributions and recognize the effectiveness of our method. Specifically, reviewers acknowledged:

- **Novel and Practical Formulation:** The redundancy-aware test-time graph OOD detection fills a gap in current data-centric, post-hoc OOD detection literature, offering a new perspective while eliminating the need for model retraining. (Reviewer mdJJ, Reviewer HHWR, Reviewer tiWp)
- **Well-Motivated Approach with Theoretical Guarantees:** The proposed ReGIB effectively eliminates redundancy and captures essential information through theoretically grounded upper and lower bound optimization. In addition, entropy minimization provides a principled and well-motivated way to extract essential information. (Reviewer mdJJ, Reviewer HHWR)
- **Strong Empirical Performance:** Extensive experiments on multiple real-world graph datasets show significant improvements over state-of-the-art baselines, supported by ablation studies and sensitivity analyses validating each component. (Reviewer mdJJ, Reviewer HHWR, Reviewer tiWp)
- **Clear and Well-Presented Work:** The manuscript is clear, well-organized, and easy to follow. (Reviewer HHWR)

We have provided detailed responses to each reviewer’s comments and have addressed all concerns. Based on the reviewers’ suggestions, we have extended our explanation of the reasons behind the limited performance on social network datasets and discussed potential solutions. We have also added further comparative analysis of time complexity, provided additional clarification on pseudo-labels, and identified adaptive or data-driven selection of coding tree height as a promising direction for future work.

Once again, we would like to express our gratitude to all the reviewers for their valuable insights, which have greatly helped us improve our work.

---

### Decision · Program_Chairs · 2025-09-17

**Decision:**

Accept (poster)

**Comment:**

The paper proposes RedOUT, a redundancy-aware framework for test-time graph Out-of-Distribution (OOD) detection. The core contribution is the introduction of the Redundancy-aware Graph Information Bottleneck (ReGIB), which decomposes graph information into essential components and redundant noise by minimizing structural entropy on a coding tree constructed from test graphs. This approach enables the extraction of distinctive substructures important for OOD detection while preserving semantic consistency. The framework operates as a post-hoc, unsupervised test-time algorithm without retraining and demonstrates substantial improvements over state-of-the-art baselines on several graph benchmarks.

**Strengths**

The strengths include its novel formulation of redundancy-aware test-time OOD detection for graphs, the introduction of ReGIB with theoretical bounds and entropy minimization, thorough ablation studies and parameter sensitivity analyses, and the framework's practicality due to not requiring model retraining. Additionally, the paper offers a new perspective on test-time graph OOD detection by addressing structural redundancy in graph data.

**Weaknesses**

The weaknesses include limited performance improvement on certain datasets (e.g., IMDB), potential error accumulation from relying on pseudo-labels during test-time, and high time complexity of coding tree construction (O(h|E|log|V|)), which may affect scalability on large graphs. Furthermore, some reviewers noted poor clarity in parts of the paper, lack of straightforward explanation for using construction loss as an OOD detection score, and vagueness in details of OOD sample detection.

**Summary**

During the rebuttal period, the authors addressed several concerns raised by reviewers, including explanations for limited performance improvement on certain datasets, impact of incorrect pseudo-labels at test time, and scalability of coding tree construction on large graphs. The authors also provided additional analyses, such as comparisons with baseline methods and discussions on selecting optimal values for different datasets. Reviewers appreciated the rebuttal, with one reviewer increasing their rating after finding that the authors had addressed most of their concerns. While some reviewers maintained their original scores, they acknowledged the authors' efforts to address concerns and recognized the paper's technical solidity despite some limitations.

Overall, while not working on an exciting new topic, the paper presents a decent contribution. Based on the reviews and rebuttal, I recommend accepting this paper. The authors have addressed most of the reviewers' concerns, providing additional explanations and analyses to clarify the methodology and results. The paper's strengths, including its novelty, thorough experiments, and good performance on benchmark datasets, outweigh its weaknesses. While some limitations remain, such as sensitivity to hyperparameters and potential computational costs, the authors have acknowledged these and plan to discuss them further in a revised version.